# Procedure-Aware Surgical Video-language Pretraining with Hierarchical Knowledge Augmentation

**Kun Yuan**[1,2,3]    **Vinkle Srivastav**[1,2]    **Nassir Navab**[3]    **Nicolas Padoy**[1,2]

[1]University of Strasbourg, CNRS, INSERM, ICube, UMR7357, Strasbourg, France
[2]IHU Strasbourg, Strasbourg, France
[3]CAMP, Technische Universität München, Munich, Germany
`{kyuan,srivastav,npadoy}@unistra.fr`
`nassir.navab@tum.de`

## Abstract

Surgical video-language pretraining (VLP) faces unique challenges due to the knowledge domain gap and the scarcity of multi-modal data. This study aims to bridge the gap by addressing issues regarding textual information loss in surgical lecture videos and the spatial-temporal challenges of surgical VLP. To tackle these issues, we propose a hierarchical *knowledge augmentation* approach and a novel Procedure-Encoded Surgical Knowledge-Augmented Video-Language Pretraining (*PeskaVLP*) framework. The proposed knowledge augmentation approach uses large language models (LLM) to refine and enrich surgical concepts, thus providing comprehensive language supervision and reducing the risk of overfitting. The PeskaVLP framework combines language supervision with visual self-supervision, constructing hard negative samples and employing a Dynamic Time Warping (DTW) based loss function to effectively comprehend the cross-modal procedural alignment. Extensive experiments on multiple public surgical scene understanding and cross-modal retrieval datasets show that our proposed method significantly improves zero-shot transferring performance and offers a generalist visual representation for further advancements in surgical scene understanding. The source code will be available at `https://github.com/CAMMA-public/PeskaVLP`.

## 1   Introduction

The recent advancements in multi-modal representation learning, particularly with the introduction of CLIP [52], have led to the development of models capable of understanding a wide range of visual concepts using natural language supervision [34, 41]. The expressive natural language has allowed these models to shift from task-specific to more generalist applications [49, 82, 83]. The learned representations of these models are robust, facilitating effective performance across diverse visual tasks without the need for task-specific fine-tuning [68, 81]. However, despite the impressive progress made by these models in the general computer vision domain, the effectiveness of these methods in domain-specific settings remains uncertain.

This concern is particularly relevant to the field of Surgical Data Science (SDS), an emerging interdisciplinary domain that utilizes deep learning and computer vision techniques to analyze surgical data [44, 43, 74]. A key component of SDS is the analysis of intraoperative surgical videos captured through endoscopes or laparoscopes. Analyzing these videos presents several unique challenges compared to the general computer vision datasets. Unlike general computer vision datasets [47, 52, 7], surgical videos can last several hours and capture complex and fine-grained activities within a narrow field of view. This requires development of computational approaches to decompose and model the surgical procedures at multiple hierarchical levels, including the entire

38th Conference on Neural Information Processing Systems (NeurIPS 2024).

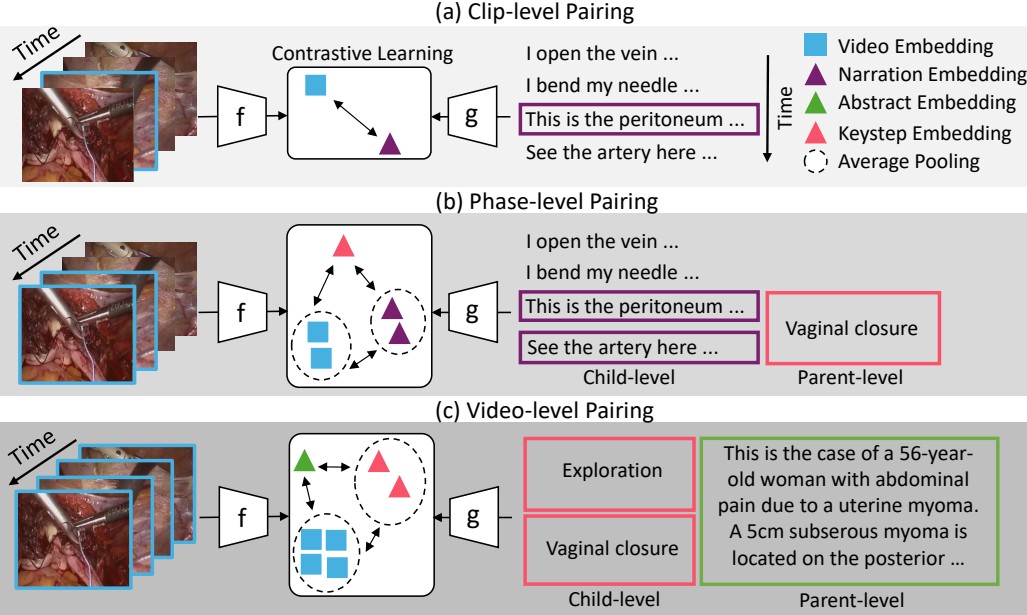

Figure 1: Illustratsion of video-language pretraining with hierarchical video-text pairs. At phase- and video-level, one parent-level text is paired to multiple child-level texts.

procedure [29], phases [67, 16], steps [54, 31], atomic actions [6, 8], and action triplets [50, 62]. Moreover, surgical language involves specialized vocabulary, and annotating videos requires clinical expertise, limiting dataset scalability. Consequently, current deep learning applications are restricted to single-centric, fully-supervised, and task-specific approaches [3, 6, 31, 50, 55, 57, 67, 69, 74].

To bridge the gap, recent efforts have focused on creating surgical video-text pretraining datasets by curating surgical lecture videos from online e-learning platforms and pairing them with transcribed narrations using audio speech recognition (ASR) methods. Subsequently, a CLIP-style model [76] is trained contrastively to match the video clips to their corresponding textual descriptions. Building on this, the HecVL approach introduces hierarchical texts, including phase-level keystep descriptions and video-level summaries that provide hierarchical goals of the surgical procedure [75]. However, challenges persist due to the smaller size of the surgical video-language pretraining dataset, noisy transcribed narrations, limited variability in phase-level descriptions, and strong temporal dependencies in surgical procedures, where actions and keysteps occur in a specific routine order. These issues hinder the accurate learning of multi-modal surgical representations.

To address these challenges, we propose **P**rocedure-**E**ncoded **S**urgical **K**nowledge-**A**ugmented **V**ideo-**L**anguage **P**retraining (PeskaVLP), which boosts data efficacy and tackles the spatial-temporal challenges inherent in surgical procedures from two perspectives. First, we introduce hierarchical knowledge augmentation to mitigate the problem of textual information loss in surgical video-language pretraining datasets. We argue that the internal knowledge of LLMs serves as a valuable surgical knowledge base, enriching and correcting text descriptions while preserving the original key concepts and meanings. Therefore, we utilize the large language model (LLM) prompted with different behaviors as an external knowledge base to correct, explain, or summarize the hierarchical texts in the surgical video-language pretraining dataset, thus providing diverse and better language supervision for multi-modal pretraining. Additionally, it reduces the risk of overfitting by preventing the text encoder from repeatedly encountering the same keystep texts in each epoch.

From the pretraining objective perspective, we perform the hierarchical video-language pretraining, as shown in Fig. 1, with a novel hierarchy-specific loss, $LecNCE$. Specifically, we combine language supervision with visual self-supervision at the clip-level pretraining to introduce additional supervision signals within vision modality, making the pretraining efficient with a small surgical dataset [76]. At phase- and video-level pretraining, we construct hard negative samples by reversing the order of texts, followed by a Dynamic Time Warping (DTW) based loss function to learn the

temporal alignment between video frames and texts, thus facilitating the understanding of cross-modal procedural alignment during pretraining.

We summarize our contributions as follows: First, we propose an LLM-based knowledge augmentation to handle surgery-specific textual information loss in the dataset, providing more densely interconnected natural language supervision from surgical lecture videos. Second, our proposed hierarchical video-language pretraining method enforces the understanding of the spatial-temporal characteristics of surgical lecture videos at different hierarchical levels. The pretrained PeskaVLP demonstrates state-of-the-art transferability and visual representation to different surgical scene understanding downstream datasets [67, 69, 31], across types of surgical procedures and clinical centers. It also shows strong multi-modal alignment ability through the cross-modal retrieval task at multiple hierarchical levels.

## 2 Related Works

**Surgical Video-Language Pretraining:** many works have demonstrated the effectiveness of learning visual representations from the natural language supervision of corresponding text [7, 70, 77, 40, 46, 42, 34]. These methods conduct contrastive learning [51] to match the video clips (or images) with their corresponding narrations (or captions). Similarly in the medical field, recent works have started to curate large-scale multi-modal data through hospital-sourced chest radiological reports [28, 12] and online platforms [76, 27, 26], e.g., YouTube and Twitter, to perform vision-language pretraining. However, these works encounter the sample efficiency issue when handling the smaller surgical video-language pretraining dataset (SVL) [76]. Recent works improve the data efficacy and zero-shot performance of CLIP-style models [48, 37, 25]. However, they do not capture procedural dependency from the long-form surgical videos beyond the video clip and text matching. Hierarchical pretraining methods [4, 79, 75] propose to pair video clips of different durations to different hierarchical levels of texts, covering both short- and long-term understanding. Paprika [80] builds a procedural knowledge graph and elicits the knowledge node during the video-language pretraining process.

**Textual Augmentation with Knowledge Base:** the success of vision-language pretraining is highly dependent on the quality and quantity of available multi-modal data. Recent research [38] shows that a smaller high-quality dataset can outperform a larger low-quality dataset. Common practices improve the quality by textual augmentation, including EDA [37], masked token modeling [65], and captioning loss [72]. Recent studies have used synthesized captions from captioning models to achieve notable improvements [33, 32, 58]. However, they show scalability deficiency and world knowledge loss in models trained with synthetic captions [73], which their initial benchmark success has largely obscured. To inject the knowledge, K-Lite [63] enriches the texts with WordNet [15] and Wiktionary [45] knowledge base. Merlot [78] learns script knowledge representations from millions of YouTube videos, however, a knowledge domain gap exists when applying this to the surgical field. The recent advent of large language models like GPT4 [2] and Llama series [66] have been a game-changer, as they encode rich domain-specific knowledge, e.g., clinical knowledge [64], motivating LaCLIP [14] to augment textual inputs through the LLM rewrites.

## 3 Approach

### 3.1 Dataset and Contrastive Learning

Learning joint video and language embedding space requires a large-scale video-language dataset, however, such datasets are expensive and time-consuming to create in the surgical field. Therefore, the first surgical video-language pretraining dataset, i.e., SVL [76], is proposed by obtaining around a thousand surgical lecture videos from surgical education platforms. SVL collects $\sim$300 hours of lecture videos accompanied by narration texts obtained using Audio Speech Recognition (ASR) methods, providing $\sim$26k video clip-narration pairs for contrastive video-language pretraining. Specifically, short video clips $x_c$ and their corresponding narration texts $y_n$ are treated as positive pairs $\mathcal{P}^n$, and the unpaired ones are treated as negative pairs $\mathcal{N}^n$. Then, the contrastive training loss

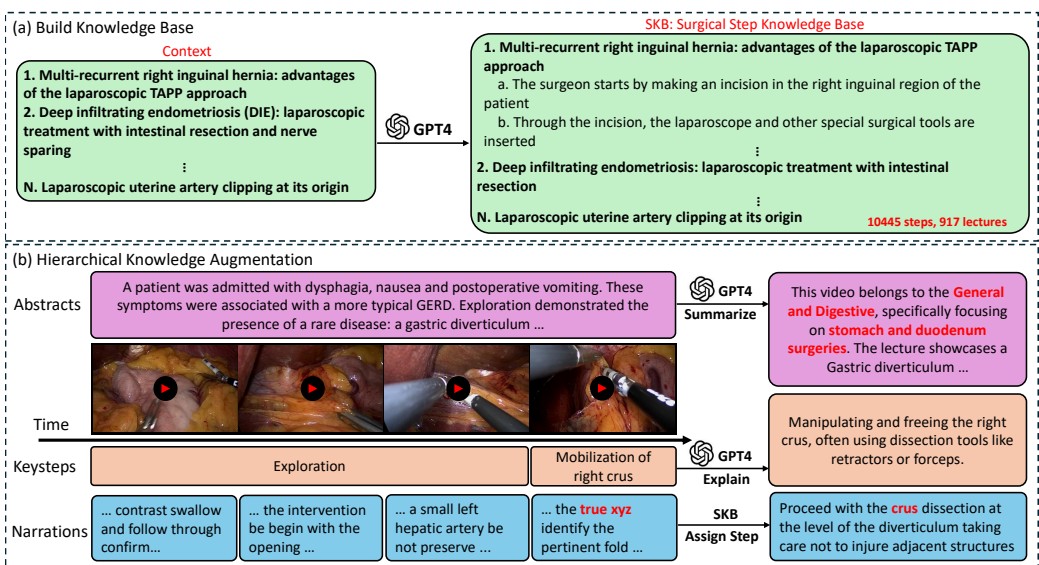

Figure 2: Hierarchical Knowledge augmentation for hierarchical texts. (a) the process of building a surgical step knowledge base. (b) the process of improving hierarchical textual quality based on LLM.

$InfoNCE$ [51] can be formulated as follows:

$$L_{InfoNCE} = \max_{f,g} \sum_{i=1}^{B} \log \left( \frac{\sum\limits_{(x_c, y_n) \in \mathcal{P}_i^n} e^{f(x_c)^\top g(y_n)}}{\sum\limits_{(x_c, y_n) \in \mathcal{P}_i^n} e^{f(x_c)^\top g(y_n)} + \sum\limits_{(x_c', y_n') \sim \mathcal{N}_i^n} e^{f(x_c')^\top g(y_n')}} \right) \tag{1}$$

where $B$ represents the batch size. The $f$ and $g$ are visual and textual encoders that generate embedding vectors for videos and texts, respectively. This loss function aligns two modalities by increasing the cosine similarity between paired videos and texts and decreasing the unpaired ones, as shown in Fig. 1 (a). Despite reaching an impressive data scale, the imprecision of the ASR system and the scarcity of surgical lecture videos limit the natural language supervision from SVL. Therefore, HecVL [75] proposes to incorporate hierarchical language supervision by extracting additional phase-level keystep and video-level abstract texts from lecture videos' metadata, as shown in Fig. 1 (b) and (c). In this work, we use this hierarchical video-language pretraining dataset and perform hierarchical knowledge augmentation to improve the textual quality.

## 3.2   Hierarchical Knowledge Augmentation

Quality of language supervision in the multi-modal representation learning matters [1, 37, 36], especially when the surgical video-language dataset is not "big" enough, e.g., millions of multi-modal samples used in [52, 47], to sufficiently cover the visual-linguistic concepts. In this work, we find that the texts suffer from different types of degradation at different hierarchies, failing to provide accurate and broad concepts for pretraining. Specifically, as shown in Fig. 2, narration texts are mostly sentence fragments and easily affected by misspelling errors, therefore altering the original key concepts. The keystep texts are mostly short and abstract, resulting in a narrow set of linguistic concepts that could show poor transferability to the downstream datasets, which usually come with a different set of concepts [63, 18]. The abstract texts sometimes include redundant and useless information, such as author and citation information.

To address the above hierarchy-specific textual degradation, we propose a hierarchical knowledge augmentation to correct, explain, and summarize the narration, the keystep, and the abstract texts, respectively, by eliciting LLM's encoded surgical knowledge [64]. For each hierarchy, we manually design the system prompt and several input-output examples for LLM. Thus, we obtain hierarchical LLM assistants with different behaviors of using internal surgical knowledge to augment the texts:

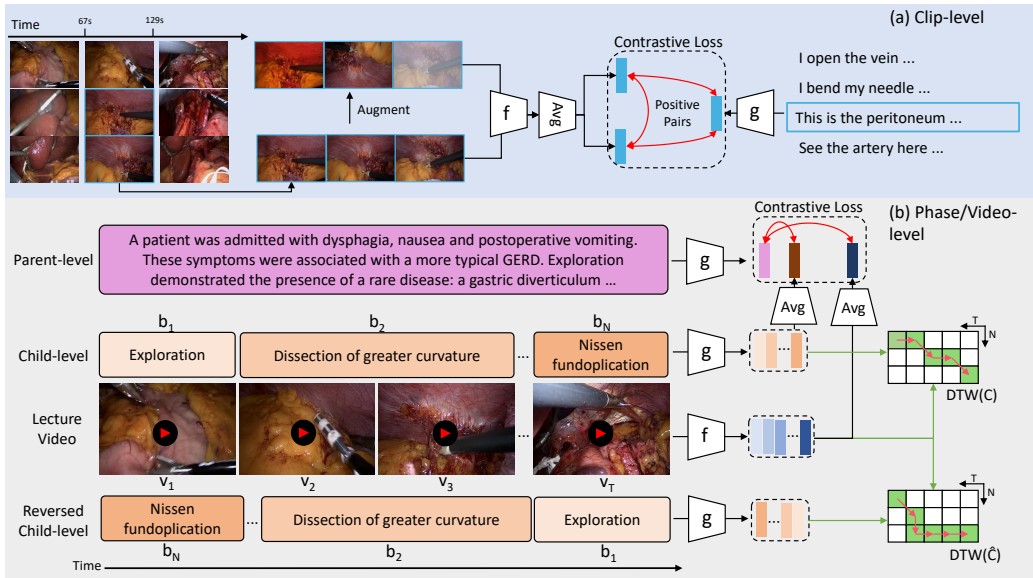

Figure 3: The pretraining pipeline of different hierarchies. We combine language supervision and visual self-supervision at clip-level pretraining. We conduct the procedure-aware contrastive learning at phase/video-level pretraining.

**Narration.** We ask the LLM to behave as a "recipe" to come up with a list of sequential steps that complete the given surgery. For each lecture video, we feed its title as input and obtain the list of pseudo steps, as shown in Fig. 2 (a), building a surgical step knowledge base. Then, we assign these pseudo steps to narration texts based on textual similarity. This implicitly corrects the typos in transcribed narrations and augments the textual input based on the LLM's surgical knowledge.

**Keystep.** As shown in Fig. 2 (b), we ask the LLM to behave like a "dictionary" to explain the meaning of the keystep. Specifically, the LLM assistant expands the given keystep into a description of the main surgical events, anatomies, and instruments involved. This enlarges the textual semantic information of each keystep and provides more expressive language supervision for pertaining.

**Abstract.** As shown in Fig. 2 (b), we ask the LLM to behave like a "summarizer" that captures the key concepts of the given abstract texts, e.g., surgical type, anatomies, and so on. This reduces the length of the textual inputs while maintaining the main concepts of the abstract paragraph. In the following experiment, we randomly input the original or augmented texts for video-language pretraining. Check Appendix H for examples of pre- and post-augmented texts.

## 3.3 Procedure-aware Surgical Video-language Pretraining

We introduce PeskaVLP, a procedure-aware pretraining framework for the above surgical knowledge-augmented video-language dataset. We emphasize devising a pretraining objective $LecNCE$ for the hierarchical video-text pairs. For clip-level pretraining, $LecNCE_{clip}$ combines language supervision with visual self-supervision to improve data efficiency and boost the scene understanding on visually similar laparoscopic images. $LecNCE_{phase/video}$ considers the procedure awareness during the coarser-level pretraining, through a DTW-based contrastive regularization objective with temporally reversed text sequences as negative samples. We apply the dual-encoder as our model architecture.

### 3.3.1 Clip-level Pretraining

**Language Supervision.** The common pretraining objective for dual-encoder model is $InfoNCE$ [51], as denoted in Eq. 1, where matched video text pairs are treated as positive while

all other pairwise combinations in the batch are regarded as negative. In this work, we also apply $InfoNCE$ to maximize the similarity between short-term video clips and their corresponding narration texts at the clip level, denoted as $L_{clip}^{vl}$. However, this simple objective is data hungry and sensitive to the weakly aligned noisy video-text pairs from small-scale surgical video-language datasets, such as SVL [76].

**Visual Self-supervision.** The proposed PeskaVLP approach introduces an additional supervision signal from visual self-supervision to complement noisy language supervision. Specifically, we explore the widespread supervision within visual modality to learn generic visual representation. We adopt the simple yet effective SimSiam [11] strategy that aims to maximize the similarity between two augmented views. As shown in Fig. 3 (a), during the pretraining, we apply random distortion on the frames of video clips and generate two augmented embedding vectors for one video clip. We then apply $InfoNCE$ to maximize the similarity of these two augmented embeddings by treating them as positive pairs, denoted as $L_{clip}^{vv}$. This additional supervisory can learn visual features more efficiently and is robust to the distortion of surgical scene images. Finally, the $LecNCE$ loss for clip-level pretraining is the sum of these two losses, denoted as $LecNCE_{clip} = L_{clip}^{vl} + L_{clip}^{vv}$.

### 3.3.2 Phase-/Video-level Pretraining

The surgical video-language pretraining presents a unique procedural challenge compared to the existing video-language methods [19, 47, 52, 71, 61]. The surgical actions and events occur in a certain order to follow the routine to complete the surgical phase and surgery, e.g., "hook dissecting cystic duct" should happen before "clipper cutting cystic duct" in the "clipping cutting" phase of cholecystectomy surgery. However, prior contrastive learning objectives [46, 52, 19] omit this temporal dependency and limit the understanding of procedural knowledge in surgical lecture videos.

Our proposed $LecNCE$ training objective enables procedural understanding in phase- and video-level pretraining by considering the cross-modal temporal alignment between video frames and text sequence. Specifically, hierarchical texts can form the parent-child correspondence, i.e., abstract (parent-level) and keystep (child-level) texts, keystep (parent-level) and narration (child-level) texts. As shown in Fig. 3 (b), each parent-level text $A$ is paired with a video segment $V = \{v_1, ...v_T\}$, where the $T$ is the number of frames of the video segment. $A$ is also paired with a child-level text sequence $B = \{b_1, ...b_N\}$, where $N$ is the length of this sequence. Then, we build the cost matrix $C \in R^{T \times N}$ between video frames and child-level text sequence based on their embeddings, with each element $c_{i,j}$ computed by a distance function $D$. We adopt the same distance function from [21]:

$$c_{i,j} = \mathcal{D}(v_i, b_j) = -\log \frac{\exp(\tilde{\mathbf{v}}_i^\top \tilde{\mathbf{b}}_j / \beta)}{\sum_{k=1}^{N} \exp(\tilde{\mathbf{v}}_i^\top \tilde{\mathbf{b}}_k / \beta)}, \quad \tilde{\mathbf{v}}_i = f(v_i)/\|f(v_i)\|_2 \quad \tilde{\mathbf{b}}_i = g(b_i)/\|g(b_i)\|_2 \tag{2}$$

Using this cost matrix $C$, we apply Dynamic Time Warping (DTW) to find the minimum cross-modal cost path that aligns the video frames to the text sequence, denoted as $DTW(C)$. We then make a reasonable assumption that the global semantics of the text sequence and its reversed version are distinct. Therefore, aligning the video frames to the text sequence should be easier, i.e., incur a lower alignment cost compared to aligning the same video frames when the text sequence is played in reverse. Following this assumption, we temporally reverse the child-level texts into $\hat{B} = \{b_n, ...b_1\}$ and build the cost matrix $\hat{C}$ between $V$ and $\hat{B}$, computing the minimum alignment cost $DTW(\hat{C})$. We then devise a DTW-based contrastive regularization using hinge loss as follows:

$$L_{dtw} = max(DTW(C) - DTW(\hat{C})), \phi) \tag{3}$$

where $\phi$ is the margin between positive and negative samples. This imposed regularization can support fine-grained multi-modal representation learning from weakly paired video frames and texts via temporal alignment. Unlike Paprika [80], which relies on a pretrained model [46], our phase-/video-level pretraining provides a direct, lightweight, and more adaptable methodology to unseen surgical domains. We do not require the adaption from any existing models, improving the generalization capability. Also, our pretraining process is procedure-aware in itself rather than modifying the representation in a second step, streamlining the process and increasing efficiency. We also apply the $InfoNCE$ loss to maximize the similarity between the paired parent-level text, video segment, and child-level texts, denoted as $L_{infonce}$. Note that the $L_{infonce}$ follows the same pipeline as in Fig. 1 (b) and (c). Finally, we achieve the loss $LecNCE$ for phase- or video-level

pretraining as $LecNCE_{phase/video} = L_{infonce} + \lambda L_{dtw}$, where $\lambda$ is the hyper-parameter to scale two losses. Please refer to Appendix D for more details about dynamic time warping. Finally, we train the model in an alternating way, using the proposed hierarchical levels of learning objectives. We only train one set of visual and textual encoders for all three levels, ensuring the encoders are optimized for capturing both short-term and long-term semantics. We alternatively train with 25 batches of clip-level samples, followed by 15 and 115 batches of phase- and video-level samples.

## 4  Experiments

**Datasets.** Our pretraining is conducted on the videos of SVL [76] dataset. The pertaining dataset includes hierarchical textual annotations from the metadata of the videos [75]. We evaluate our model on 3 publicly available surgical phase recognition downstream datasets, i.e., Cholec80 [67] (cholecystectomy) from Strasbourg center, AutoLaparo [69] (hysterectomy) from HongKong hospital, MultiBypass140 [31] (gastric bypass) from both Strasbourg (StrasBypass70) and Bern (BernBypass70) centers. These datasets contain untrimmed surgical workflows with frame-wise phase labels. We also evaluate pretrained model on the cross-modal retrieval task in multiple hierarchical levels with holdout videos in SVL-Retrieval [76]. Check Appendix A for more details about pretraining dataset.

**Training Parameters.** We utilize the dual-encoder architecture with ResNet50 [23] as visual encoder and ClinicalBert [24] as textual encoder, respectively. We train the model with a batch size of 120/80/25 for clip-/phase-/video-level, respectively. We sample 4/16/64 frames for videos of clip-/phase-/video-level. We use AdamW optimizer [30] with a learning rate of $5e-5$. We train the model with 4 NVIDIA A100 GPUs each having a DRAM of 80 GB for 200 epochs. Temperature parameter $\beta$ for distance function and $\phi$ for DTW-base contrastive loss function $D$ are fixed as 0.1. Scale factor $\lambda$ is set as 0.01.

**Evaluation Setup.** We evaluate pretrained models using two setups: Zero-Shot evaluation and Few/Full-shot Linear Probing evaluation. For Zero-Shot, we utilize class text prompts, the same as HecVL [75], to compute cosine similarities between image embedding and class text embeddings, classifying images based on the shortest distance. In Linear Probing, the pretrained visual encoder remains frozen when we extract features for each image, subsequently training a linear layer using the SGD optimizer. For few-shot linear probing, we train the linear layer with a few numbers of videos, referred to as $k$-% training, where $k$ indicates the percentage of all the videos used in training. Check Appendix B for more details.

Table 1: Zero-shot phase recognition results. We report Accuracy / F1-Score. PeskaVLP outperforms the other methods across different tasks. We report the state-of-the-art methods that are fine-tuned on the downstream dataset in a fully-supervised manner. However, models fine-tuned on specific downstream datasets show limited generalizability across procedures and institutions.

| Model | Dataset | Cholec80 | Autolaparo | StrasBypass70 | BernBypass70 | Average |
|---|---|---|---|---|---|---|
| TransVNet [17] | Cholec80 | 90.3 / – | – / – | – / – | – / – | – / – |
| | Autolaparo | – / – | 82.0 / – | – / – | – / – | – / – |
| ResNet50 [31] | BernBypass | – / – | – / – | 57.3 / 32.7 | 85.3 / 62.4 | – / – |
| | StrasBypass | – / – | – / – | 90.2 / 79.9 | 56.7 / 29.5 | – / – |
| MIL-NCE [46] | Howto100M | 7.8 / 7.3 | 9.9 / 7.9 | 5.6 / 3.1 | 2.4 / 2.1 | 6.4 / 5.1 |
| CLIP [52] | CLIP400M | 30.8 / 13.1 | 17.4 / 9.1 | 16.9 / 5.5 | 14.8 / 4.1 | 19.9 / 8.0 |
| | Scratch | 29.4 / 10.4 | 15.3 / 10.9 | 6.3 / 3.5 | 4.9 / 2.3 | 14.0 / 6.8 |
| | SVL | 33.8 / 19.6 | 18.9 / 16.2 | 15.8 / 8.6 | 17.8 / 7.1 | 21.6 / 12.9 |
| SurgVLP [76] | SVL | 34.7 / 24.4 | 21.3 / 16.6 | 10.8 / 6.9 | 11.4 / 7.2 | 19.6 / 13.8 |
| HecVL [75] | SVL | 41.7 / 26.3 | 23.3 / 18.9 | 26.9 / 18.3 | 22.8 / 13.6 | 28.7 / 19.3 |
| PeskaVLP | SVL | **45.1 / 34.2** | **26.5 / 23.6** | **46.7 / 28.6** | **45.7 / 22.6** | **41.0 / 27.1** |

### 4.1  Zero-shot Surgical Phase Recognition

**High-quality Surgical Video-language Dataset.** As shown in Table 1, our approach achieves a significant performance improvement over the baselines MIL-NCE [46] and CLIP [52] pretrained on the natural computer vision datasets, even though our pretraining dataset is $10,000$ times smaller

than those. Note that when the CLIP model is randomly initialized and then trained with SVL, its performance declines compared to initializing from OpenAI. This shows that our surgical video-language pretraining dataset lacks the scale necessary to adequately pretrain a robust video-language model from scratch. ViT [13, 9] architectures are sensitive to initialization and excluded from this work. Further insights into the impact of initialization can be found in Appendix C.

**Transferability across Surgical Procedures and Centers.** Compared to the HecVL, our method achieves over 12.3% and 7.8% improvement in absolute accuracy and f1, thanks to our spatial-temporal $LecNCE$ learning objective across multiple hierarchies. Also, the consistent boost on cholecystectomy [67], hysterectomy [69], and gastric bypass [**?** ] procedures show the generalizable and transferable features of PeskaVLP. Comparing the results of StrasBypass and BernBypass, we find that PeskaVLP can recognize the phases of the same kind of surgery (gastric bypass), even if these surgeries are performed in different centers and follow different procedural routines. More qualitative results can be found in Appendix F.

## 4.2 Zero-shot Cross-modal Retrieval

Table 2: We present cross-modal retrieval results on the holdout videos, highlighting the best performance in each setting in bold. We additionally include coarser-grained phase-keystep and abstract-video text pairs to assess long-term video and high-level textual understanding.

| method | Clip-Narration | | | Phase-Keystep | | | Video-Abstract | | |
|---|---|---|---|---|---|---|---|---|---|
| | R@1 | R@5 | R@10 | R@1 | R@5 | R@10 | R@1 | R@5 | R@10 |
| | Text-to-Image (%) | | | | | | | | |
| CLIP [52] | 2.9 | 5.2 | 6.7 | 1.7 | 3.2 | 6.3 | 1.2 | 11.7 | 25.8 |
| SurgVLP [76] | 2.8 | 11.8 | 16.1 | 1.6 | 6.8 | 11.6 | 1.3 | 8.2 | 15.5 |
| HecVL [75] | 2.7 | 11.3 | 17.2 | 3.9 | 13.7 | 21.3 | 28.2 | 74.1 | 82.3 |
| PeskaVLP | **3.2** | **13.2** | **23.3** | **6.1** | **21.0** | **35.4** | **38.8** | **75.3** | **85.9** |
| | Image-to-Text (%) | | | | | | | | |
| CLIP [52] | 1.8 | 3.9 | 6.0 | 0.3 | 1.2 | 2.7 | 0 | 7.0 | 16.4 |
| SurgVLP [76] | 1.3 | 8.6 | 13.5 | 1.0 | 4.1 | 7.3 | 1.3 | 8.6 | 14.6 |
| HecVL [75] | 2.1 | 9.0 | 16.2 | 1.9 | 8.3 | 14.8 | 21.2 | 65.9 | 71.8 |
| PeskaVLP | **2.4** | **13.1** | **21.3** | **3.4** | **14.9** | **24.8** | **38.8** | **75.3** | **81.1** |

In our study, we evaluate pretrained models' cross-modal alignment efficacy by conducting both zero-shot text-to-image and image-to-text retrieval tasks in multiple hierarchical levels. We report the Recall@N metric by identifying the retrieved nearest neighbors for each query and then determining whether the corresponding ground truth element is within the top $N$ nearest neighbors, where $N \in \{1, 5, 10\}$. Table 2 shows that our PeskaVLP achieves superior performance due to the procedure-aware learning objective in hierarchical pretraining. Particularly, the hierarchical pretraining scheme significantly boosts the cross-modal retrieval at the coarse-grained video-text pairs, comprehending the relationship between long video segments and high-level sentences with surgical terms.

## 4.3 Few-/Full-shot Linear Probing

**General Visual Representation for Surgical Scene Understanding.** We present the few- and full-shot linear-probing evaluation in Table 3. It shows that the learned visual representation from PeskaVLP provides a general visual representation for surgical scene understanding across surgical procedures. We also find that the MoCo v2 [55, 22] pretrained on the frames of the SVL dataset (second row of Table 3) in a visual self-supervised manner achieves better visual representation than pretraining on a public dataset that only contains one type of surgery, e.g., Cholec80 (third row in Table 3). This shows that the cross-procedure surgical pretraining dataset enables better generalizationability.

**Knowledge Augmentation and Hierarchical Pretraining.** Interestingly, the model pretrained contrastively with short video clips and narrations (SurgVLP) performs worse than MoCo v2 [55, 22] (second row in Table 3) in linear probing evaluation. This may be because the noisy narrations do not provide accurate natural language supervision for visual representation learning, thus highlighting the

Table 3: Linear-probing evaluation results. V: supervision is from visual frames. L: supervision is from natural languages. VL: supervision is from both visual and language entities.

| Model | Dataset | k-% | Cholec80 | Autolaparo | StrasBypass70 | BernBypass70 |
|---|---|---|---|---|---|---|
| ImageNet | ImageNet (V) | 100 | 66.4 / 54.9 | 57.5 / 44.9 | 66.2 / 53.6 | 64.7 / 31.6 |
| | | 10 | 57.4 / 42.3 | 44.9 / 30.4 | 53.3 / 42.1 | 53.3 / 25.6 |
| MoCo v2 [55] | SVL (V) | 100 | 68.2 / 55.8 | 59.5 / 48.4 | **71.6** / 58.1 | 69.6 / 36.5 |
| | | 10 | 57.6 / 43.5 | 49.9 / 34.6 | 63.1 / 49.3 | 59.1 / 29.9 |
| MoCo v2 [55] | Cholec80 (V) | 100 | **73.4 / 62.8** | 51.3 / 37.4 | 67.8 / 55.4 | 66.0 / 33.1 |
| | | 10 | **69.6 / 56.9** | 45.4 / 31.7 | 58.1 / 45.2 | 52.7 / 25.7 |
| CLIP [52] | NA (L) | 100 | 64.8 / 50.7 | 58.5 / 46.1 | 65.4 / 50.6 | 64.1 / 33.3 |
| | | 10 | 57.5 / 40.0 | 46.2 / 31.4 | 54.3 / 42.1 | 52.8 / 27.9 |
| CLIP [52] | SVL (L) | 100 | 64.9 / 55.0 | 53.1 / 42.1 | 69.1 / 55.7 | 68.2 / 35.2 |
| | | 10 | 58.9 / 42.3 | 45.3 / 35.3 | 58.2 / 45.2 | 56.5 / 29.8 |
| SurgVLP [76] | SVL (L) | 100 | 63.5 / 50.3 | 54.3 / 41.8 | 65.8 / 50.0 | 66.5 / 34.3 |
| | | 10 | 55.0 / 39.9 | 48.5 / 32.0 | 57.0 / 44.0 | 57.7 / 28.5 |
| HecVL [75] | SVL (L) | 100 | 66.0 / 53.2 | 56.9 / 44.2 | 69.8 / 54.9 | 70.0 / 34.4 |
| | | 10 | 56.1 / 40.3 | 46.9 / 32.1 | 60.2 / 46.8 | 59.3 / 31.2 |
| PeskaVLP | SVL (VL) | 100 | 69.9 / 59.8 | **63.1** / **49.7** | 71.4 / **59.5** | **71.5 / 37.4** |
| | | 10 | 61.9 / 50.6 | **53.1 / 36.8** | **63.8 / 50.4** | **62.9 / 32.7** |

Table 4: Ablation study on different modifications. Knowledge: knowledge augmentation applied to the pretraining dataset at phase-level (P) and video-level texts (V). P/V: procedure-aware pretraining learning objective at phase and video-level. C: the integration of language and visual self-supervision at clip-level pretraining. We report 10%-shot linear probing in this table.

| LecNCE | | Knowledge | | Zero-shot | | Linear-probing | |
|---|---|---|---|---|---|---|---|
| P/V | C | P | V | Cholec80 | Autolaparo | Cholec80 | Autolaparo |
| ✗ | ✗ | ✗ | ✗ | 41.7 / 26.3 | 23.3 / 18.9 | 56.1 / 40.3 | 46.9 / 32.1 |
| ✗ | ✓ | ✗ | ✗ | **45.5** / 31.0 | 25.3 / 20.0 | – / – | – / – |
| ✗ | ✗ | ✓ | ✓ | 42.4 / 28.1 | 24.9 / 20.4 | 58.1 / 43.2 | 48.5 / 34.7 |
| ✗ | ✓ | ✓ | ✓ | 43.4 / 30.3 | **28.3 / 24.5** | 60.4 / 48.6 | **53.8 / 39.2** |
| ✓ | ✓ | ✓ | ✗ | 44.0 / 31.8 | – / – | – / – | – / – |
| ✓ | ✓ | ✗ | ✓ | 43.7 / 30.6 | – / – | – / – | – / – |
| ✓ | ✓ | ✓ | ✓ | 45.1 / **34.2** | 26.5 / 23.6 | **61.9 / 50.6** | 53.1 / 36.8 |
| | | | | StrasBypass70 | BernBypass70 | StrasBypass70 | BernBypass70 |
| ✗ | ✗ | ✓ | ✓ | 26.9 / 18.3 | 22.8 / 13.6 | 60.2 / 46.8 | 59.3 / 31.2 |
| ✗ | ✗ | ✓ | ✓ | 32.3 / 21.2 | 23.8 / 17.5 | 62.6 / 47.7 | 60.3 / 32.3 |
| ✗ | ✓ | ✓ | ✓ | 39.8 / 23.7 | 25.7 / 21.3 | 63.5 / 48.6 | 62.2 / 32.0 |
| ✓ | ✓ | ✓ | ✓ | **45.1 / 34.2** | **26.5 / 23.6** | **63.8 / 50.4** | **62.9 / 32.7** |

importance of visual self-supervision and textual quality. Our model surpasses the prior methods by a large margin, showing the efficacy of our hierarchical knowledge augmentation, which denoises the text and improves textual quality. Also, our proposed $LecNCE$ promotes the visual encoder through additional visual self-supervision and procedural understanding. We present t-SNE visualizations of learned features in Appendix E, which shows that our multi-modal representations exhibit a smaller modality gap, enhancing transferability to vision-and-language downstream tasks [20, 39].

## 4.4 Ablation Studies

**Effect of Knowledge Augmentation.** Table 4 presents the effect of our proposed LLM-based hierarchical knowledge-aware augmentation strategy, applied to the texts of SVL dataset. The first row of the table corresponds to HecVL [75] pretrained on SVL with only conventional visual augmentations, e.g., blurring and so on, without any knowledge augmentation. The results clearly demonstrate that simple visual augmentation strategies exhibit poor robustness as the texts of SVL are noisy and not diverse enough. Conversely, our knowledge-aware text augmentation consistently improves performance across multiple surgical datasets, highlighting the importance of the textual quality of

the surgical video-language pretraining dataset. We found that integrating visual self-supervision with language supervision significantly enhances performance in surgical scene understanding tasks across downstream datasets. Additionally, using a procedure-aware learning objective improves surgical phase recognition for routine procedures, such as cholecystectomy (Cholec80), more effectively than complex procedures, like hysterectomy (Autolaparo).

**Effect of Pretraining Objective.** Table 4 shows the impact of our learning objective for hierarchical surgical video-language pretraining. When we append visual self-supervision to language supervision at the clip-level pretraining, the zero-shot performance is clearly improved. This improvement can be attributed to the added diverse and high-quality supervision. Also, the boost at linear-probing evaluation shows that the combination of language supervision and visual self-supervision leads to a robust visual representation especially with a moderate size of surgical video-language dataset, e.g., SVL. Table 4 also highlights that the inclusion of $LecNCE$ with procedure understanding consistently improves performance across most downstream datasets, leading to enhanced accuracy in both zero-shot and linear-probing. However, performance on the AutoLaparo degrades with this modification. This may be due to challenging or less routined surgical procedures in the pretraining dataset.

## 5   Conclusion, Limitations and Broader Impact

**Conclusion.** We have introduced a surgical video-language pretraining method for long-term surgical lecture videos and their hierarchical paired texts. Our proposed knowledge augmentation addresses the hierarchical textual information loss by integrating the large language model's internal surgical knowledge. Also, we propose a novel spatial-temporal pretraining objective for video-text pairs of different hierarchies, which addresses the lack of supervision signals problem in a small surgical vision-language dataset. The proposed $LecNCE$ also addresses the procedural awareness problem, benefiting the long-term cross-modal understanding. The experiments show that our proposed PeskaVLP achieves the state-of-the-art generalized zero-shot ability and visual representation learning that can serve as a general initialization for many surgical scene understanding tasks.

**Limitations.** While our LLM-augmented strategy enhances textual information, it may overly standardize the text, raising concerns about overfitting during pretraining. Therefore, it is crucial to strike a balance between leveraging LLM capabilities and maintaining the variability present in real-world surgical narratives. To address this, future work will explore incorporating diverse audio inputs and spontaneous narratives into the pretraining process, ensuring that the model retains robustness and adaptability in real-world applications. Additionally, even though the SVL pretraining dataset covers diverse laparoscopic surgeries, it lacks surgeries in different organs, such as the brain and heart. To address this, we plan to expand the pretraining dataset using diverse media such as textbooks, instructional videos, and intraoperative video recordings from diverse sources. We also aim to diversify the pretraining dataset by considering laparoscopic, endoscopic, and microscopic surgeries on different organs, to further mitigate the risk of overfitting and enhance the model's generalizability.

**Broader Impact.** The primary goal of surgical data science is to develop novel context-aware support systems for the operating room by collecting large-scale surgical data and analyzing it with modern AI techniques, eventually improving the safety and efficacy of surgical outcomes. The recent advancements in vision-language-based multi-modal AI offer significant potential in achieving this goal by enabling the development of more robust and generalizable models. These multi-modal systems have the potential to support clinical decision-making, streamline surgical workflows, provide real-time intra-operative guidance to improve surgical precision, reduce errors, and optimize outcomes in the operating room. During the development, patient data privacy should be considered as a fundamental ethical requirement. These systems developed on real-world surgical data also hold transformative potential in medical education, enhancing training and skill development in both novice and experienced surgeons.

## Acknowledgements

We would like to extend our deep appreciation to the education platforms, such as Websurg (IRCAD), EAES, and YouTube, for their dedication to providing high-quality educational content freely accessible to learners worldwide. We are especially grateful to the clinicians who have generously contributed their time and expertise to create and share content on these platforms, making this research possible.

This work has received funding from the European Union (ERC, CompSURG, 101088553). Views and opinions expressed are however those of the authors only and do not necessarily reflect those of the European Union or the European Research Council. Neither the European Union nor the granting authority can be held responsible for them. This work was also partially supported by French state funds managed by the ANR under Grants ANR-20-CHIA-0029-01 and ANR-10-IAHU-02. This work was granted access to the HPC resources of IDRIS under the allocations AD011013704R1, AD011011631R2, and AD011011631R4 made by GENCI. The authors would like to acknowledge the High-Performance Computing Center of the University of Strasbourg for supporting this work by providing scientific support and access to computing resources. Part of the computing resources were funded by the Equipex Equip@Meso project (Programme Investissements d'Avenir) and the CPER Alsacalcul/Big Data.

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

# A Pretraining Dataset

## A.1 Videos

We start with the videos that are used for surgical vision-language pretraining in [76]. In total, there are $1,326$ surgical lecture videos. These videos are transcribed by AWS [5] and Whisper [53] audio speech recognition (ASR) to obtain the corresponding narration texts. Furthermore, we curate the videos' metadata from the online platforms to obtain the extra keystep and abstract texts. In the phase- and video-level pretraining, we need parent- and child-level text correspondences, e.g., keystep and its corresponding narration texts, to perform procedure understanding. Therefore, we filter out the videos that do not have parent-child correspondences. In total, we have $1,007$ and $920$ videos for phase- and video-level pretraining, respectively.

## A.2 Misspelling Error

As the narration texts are generated from the audio using the ASR system, they usually contain many misspelling errors and fragment sentences. Therefore, we apply multiple preprocessing steps to clean the narration texts.

We first built the vocabulary based on the textbook, surgical category labels, and definition words. Specifically, we refer to the academic papers, which define the surgical phases, to curate a list of definition words and build a vocabulary that contains the words of interest. We also parse and merge the words from the textbook. In total, we obtain a vocabulary of the size of $51,640$ words. Then, we use the built vocabulary along with the spell-checking algorithm [1] to correct the misspelling errors in narration texts. The algorithm utilizes Levenshtein Distance to identify words within 2 edit distances from the original. It then cross-references these permutations (insertions, deletions, replacements, and transpositions) with a word frequency list, prioritizing words with higher occurrence frequencies as potential correct results.

# B Evaluation Setup

We provide a detailed description of the downstream tasks and their settings that we apply in the experiment.

**Surgical Phase Recognition.** Surgical phase recognition is a proxy task to test the model's surgical scene understanding ability. It aims to classify the frame of surgical video into predefined classes (phases), requiring the model to understand the instrument and anatomy's presence and their interactions by extracting visual patterns from the surgical scene image. In this work, we ignore temporal modeling in surgical phase recognition as we focus on multi-modal representation learning. We consider phase recognition as a frame-wise image classification problem. In the surgical phase recognition task, we evaluate the model's performance based on the publicly available datasets, including Cholec80 [67], AutoLaparo [69] and MultiBypass [**?** ].

- **Zero-shot Evaluation.** As the surgical phase labels are high-level definitions that can be decomposed into a few basic concepts, we manually construct the contextual prompts for phase labels, as shown in Tab. 5, Tab. 6 and Tab. 7. Our constructed prompts for the class names are built with the help of clinician's comments, considering the involved surgical instruments, anatomies, and events involved in a given surgical phase.

- **Linear-probing Evaluation.** For linear-probing evaluation on the surgical phase recognition downstream datasets, we keep the visual encoder frozen and train a linear classifier on the extracted features. We do not apply any image augmentation during the training. The learning rate is scaled linearly based on the actual batch size. The model is optimized using SGD optimizer with the learning rate as $0.001$ and weight decay parameter as $0.0005$. We train the model for 40 epochs. We fit the model on the training and validation sets and report the performance on the separate test set. For the few-shot linear-probing evaluation, we adopt a $k$-percentage shot approach with a slight modification to accommodate the nature of surgical videos, which contain frames from different classes. Specifically, we select 10%

---

[1]https://github.com/barrust/pyspellchecker/

Table 5: Manually designed prompts for the class names to recognize the surgical phase in Cholec80 dataset. We decompose high-level phase definitions into a few basic concepts to form the text prompts.

| Phase Labels | Prompts |
| --- | --- |
| *Preparation* | In preparation phase I insert trocars to patient abdomen cavity |
| *CalotTriangleDissection* | In calot triangle dissection phase I use grasper to hold gallbladder and use hook to expose the hepatic triangle area and cystic duct and cystic artery |
| *ClippingCutting* | In clip and cut phase I use clipper to clip the cystic duct and artery then use scissor to cut them |
| *GallbladderDissection* | In dissection phase I use the hook to dissect the connective tissue between gallbladder and liver |
| *GallbladderPacking* | In packaging phase I put the gallbladder  into the specimen bag |
| *CleaningCoagulation* | In clean and coagulation phase I use suction and irrigation to clear the surgical field and coagulate bleeding vessels |
| *GallbladderRetraction* | In retraction phase I grasp the specimen bag and remove it from trocar |

Table 6: Manually designed prompts for the class names to recognize the surgical phase in AutoLaparo dataset.

| Phase Labels | Prompts |
| --- | --- |
| *Preparation* | I use grasper to grasp and explore the field |
| *Dividing Ligament and Peritoneum* | I divide ligament and peritoneum |
| *Dividing Uterine Vessels and Ligament* | I divide uterine vessels and ligament |
| *Transecting the Vagina* | I use the dissecting hook to transect the vagina |
| *Specimen Removal* | I remove the specimen bag and uterus |
| *Suturing* | I suture the tissue |
| *Washing* | Washing |

of the video from the training set. This ensures that data leakage is prevented and that the number of samples per class remains similar.

**Cross-modal Retrieval.** Cross-modal retrieval includes text-based video retrieval and video-based text retrieval. Here, we conduct the cross-modal retrieval at three hierarchical levels. We collect 537 clip-narration (clip-level) video-text pairs, 746 phase-keystep (phase-level) video-text pairs, and 86 video-abstract (video-level) video-text pairs from hold-out testing videos of SVL [76]. There are more phase-keystep than clip-narration video-text pairs because some testing videos do not have cleaned narrations and we filter them out. For video embedding generation, we sample multiple frames fro m the video and average pool their image embeddings. We temporally sample 10 frames for clip-/phase-/video-level videos. We conduct the zero-shot evaluation for the cross-modal retrieval task.

## C    Architecture & Initialization

As mentioned before, the current surgical vision-language pretraining dataset lacks the scale necessary to pretrain a robust vision-language model from scratch, therefore a good choice of architecture and initialization is important. In this section, we conduct the experiment and study the effect of different model architectures and initializations, justifying our choice of using ResNet50 architecture with ImageNet initialization as our starting point before the video-language pretraining.

- ResNet50. For ImageNet initialization, we use public IMAGENET1K_V1 weights from torchvision. Random initialization means that we random initialize the visual encoder before the hierarchical vision-language pretraining. These models' textual encoders are initialized from BioClinicalBert [24]. For CLIP initialization, we initialize the visual and textual encoder from OpenAI's weight [52].

Table 7: Manually designed prompts for the class names to recognize the surgical phase in gastric bypass dataset. We use the same prompts for both StrasBypass70 and BernBypass70. We exclude the "other" class as its definition is ambiguous.

| Phase Labels | Prompts |
|---|---|
| *Preparation* | In preparation phase I insert trocars to the abdominal cavity and expose of the operating field |
| *Gastric pouch creation* | I cut the fat tissue and open retrogastric window at stomach |
| *Omentum division* | I grasp and lift the omentum and divide it |
| *Gastrojejunal anastomosis* | I see the proximal jejunum and determine the length of the biliary limb. I open the distal jejunum and create the gastrojejunostomy using a stapler. I reinforcement of the gastrojejunostomy with an additional suture. |
| *Anastomosis test* | I place the retractor and move the gastric tube and detect any leakage of the gastrojejunostomy |
| *Jejunal separation* | I open the mesentery to facilitate the introduction of the stapler and transect the jejunum proximal |
| *Petersen space closure* | I expose between the alimentary limb and the transverse colon and close it with sutures |
| *Jejunojejunal anastomosis* | I expose between the alimentary limb and the transverse colon and close it with sutures |
| *Mesenteric defect closure* | I expose the mesenteric defect and then close it by stitches |
| *Cleaning and coagulation* | In clean and coagulation phase I use suction and irrigation to clear the surgical field and coagulate bleeding vessels |
| *Disassembling* | I remove the instruments, retractor, ports, and camera |

| Backbone | Init. | Zero-shot | | Linear-probing (10-shot) | | Linear-probing (full-shot) | |
|---|---|---|---|---|---|---|---|
| | | **Cholec80** | **Autolaparo** | **Cholec80** | **Autolaparo** | **Cholec80** | **Autolaparo** |
| ResNet50 | Random | 29.4 / 10.4 | 15.3 / 10.9 | 42.4 / 22.1 | 33.4 / 20.2 | 44.6 / 25.3 | 30.7 / 19.3 |
| | ImageNet | 34.7 / 24.4 | 21.3 / 16.6 | 55.0 / 39.9 | 48.5 / 32.0 | 63.5 / 50.3 | 54.3 / 41.8 |
| | CLIP | 33.8 / 19.6 | 18.9 / 16.2 | 58.9 / 42.3 | 45.3 / 35.3 | 64.9 / 55.0 | 53.1 / 42.1 |
| ViT-B/16 | Random | 20.2 / 11.5 | 9.1 / 8.3 | 38.4 / 20.9 | 32.1 / 19.7 | 48.2 / 25.9 | 38.4 / 25.5 |
| | ImageNet | 42.8 / 25.1 | 20.5 / 15.5 | 57.4 / 40.5 | 47.8 / 31.9 | 60.6 / 48.9 | 56.3 / 44.5 |
| | Dino | 35.1 / 19.1 | 13.9 / 9.2 | 54.7 / 39.2 | 47.4 / 31.1 | 64.9 / 51.2 | 54.0 / 42.4 |

Table 8: The experiments show that the initialization largely influences the performance of surgical video-language pretraining.

- ViT-B/16. For ImageNet initialization, we use weights from the official Google JAX implementation, which is pretrained on ImageNet21k [56] and then finetune on ImageNet1k [59]. We use the public pretrained weights from [10] for Dino initialization.

In our work, we choose ResNet50 over Vision Transformer (ViT-B/16) due to its superior performance and lower parameter amounts in the context of video-language pretraining for surgical data. Our experiments demonstrated that ResNet50, particularly when initialized with CLIP weights, outperformed ViT-B/16 across various tasks, including zero-shot and linear-probing evaluations on Cholec80 and Autolaparo datasets. Despite the advanced capabilities of vision transformers, their performance heavily depends on large-scale pretraining datasets, which might not always be available or optimal for specialized domains like surgical scenes. Conversely, convolutional neural networks like ResNet50 have shown robust generalization abilities, even when pretrained on natural images, making them more suitable for our specific application. Additionally, the initialization sensitivity observed in ViT-B/16 further justified our preference for ResNet50, ensuring a more reliable and effective starting point for our hierarchical vision-language pretraining.

# D Dynamic Time Warping

After achieving the cost matrix $C$ and $\hat{C}$, we perform dynamic time warping (DTW) [60] to find the minimum cost path to align the frames of video segment $V = \{v_1, ...v_T\}$ to the text sequence $B = \{b_1, ...b_N\}$ and reversed text sequence $\{b_N, ...b_1\}$, respectively, as shown in Algorithm. 1. We follow [71] to process the DTW function into differentiable, enabling the gradient back-propagation. The differentiable loss function is the same as [21].

A significant advantage of using DTW is that it does not require additional temporal modules, such as recurrent neural networks or attention mechanisms, to model temporal relationships. This simplification allows us to focus on learning better representations by directly aligning video frames and text sequences based on their semantics.

---

**Algorithm 1** DTW to align sequences using cost matrix

---
1: **procedure** ALIGNSEQUENCES($C, V, B$)
2:     Let $T$ be the length of sequence $V$ and $N$ be the length of sequence $B$.
3:     Set $i$ to $T$ and $j$ to $N$.
4:     Initialize $distance$ to 0.
5:     **while** $i > 0$ and $j > 0$ **do**
6:         $distance = distance + C[i][j]$
7:         **if** $i > 1$ and $j > 1$ and $C[i-1][j-1] \leq C[i-1][j]$ **and** $C[i-1][j-1] \leq C[i][j-1]$ **then**
8:             $i \leftarrow i - 1$
9:             $j \leftarrow j - 1$
10:         **else if** $i > 1$ and $C[i-1][j] \leq C[i][j-1]$ **then**
11:             $i \leftarrow i - 1$
12:         **else**
13:             $j \leftarrow j - 1$
14:         **end if**
15:     **end while**
16:     **return** $distance$.
17: **end procedure**

---

# E Modality Gap

Modality gap is a geometric phenomenon observed in the embedding space of multi-modal models [39]. This gap illustrates that pretrained multi-modal (vision-language) models create a joint embedding space where different modalities, such as images and text, are kept at a significant distance from each other. During contrastive optimization, this separation created at initialization is maintained to the extent that irrelevant image embeddings can be closer to each other than to their corresponding relevant text embeddings. This spatial disparity in the embedding space hinders the model's ability to effectively align and understand the relationships between visual and textual data, leading to suboptimal performance in tasks requiring integrated multi-modal comprehension. The existence of the modality gap is particularly detrimental when adapting pretrained vision-language models to cross-modal generation tasks, such as image captioning. As highlighted by several studies [35, 20], narrowing modality gap correlates with improved performance in cross-modal tasks.

As shown in Fig. 4, we visualize the embeddings of videos and their corresponding text descriptions at three hierarchical levels: clip-narration, phase-keystep, and video-abstract. Our proposed model demonstrates a significant reduction in the modality gap compared to the SurgVLP model. This alignment across different hierarchical levels ensures a more comprehensive and cohesive understanding of the multi-modal data, leading to superior performance in tasks like image captioning and other vision-language applications.

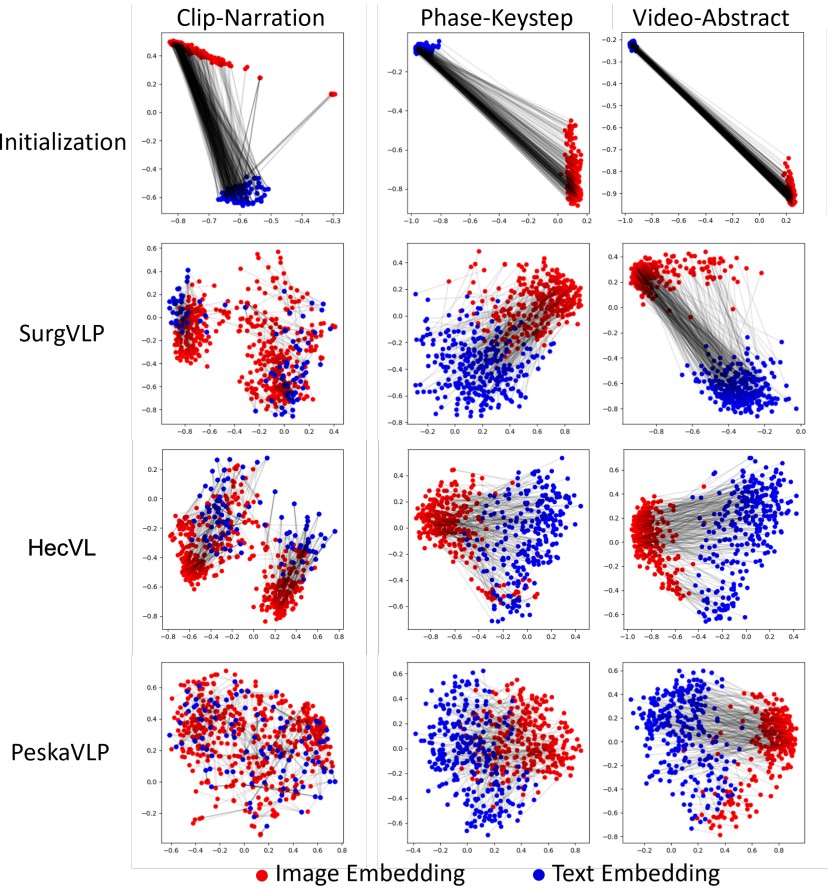

Figure 4: Modality gap visualization in different hierarchical levels. It shows that our model closes the modality gap incurred from the initialization after the hierarchical pretraining.

## F   Surgical Phase Recognition Results

We demonstrate the zero-shot surgical phase recognition to reflect the surgical scene understanding ability of our pretrained model. Our model can identify surgical phases of different types of surgical procedures without any finetuning. Both success and failure examples are shown.

**Surgical Term Understanding.** In Fig. 5, we show that the pretrained model excels at identifying the "washing" phase in surgical procedures, demonstrating its capability to accurately recognize high-level surgical activities. This proficiency enhances surgical assistance systems, improving real-time analysis and decision-making in operating rooms.

**Instrument Identification.** In Fig. 6, we demonstrate how the visual embedding is significantly influenced by the presence of surgical instruments. Specifically, in the first row, the semantic meaning of the image changes from "calot triangle dissection" to "clip and cut" due to the appearance of a hook, even though the other anatomical features remain similar.

## G   Limitations

As the pretraining process at clip-level requires additional supervision signals, i.e., visual self-supervision, the memory and computation overhead increase compared to the vanilla HecVL pretraining. Also, during the phase- and video-level pretraining, the process of dynamic time warping can be time-consuming because it is based on dynamic programming, slowing down the pretraining iteration when handling longer-term surgical videos. Additionally, the knowledge augmentation on keystep

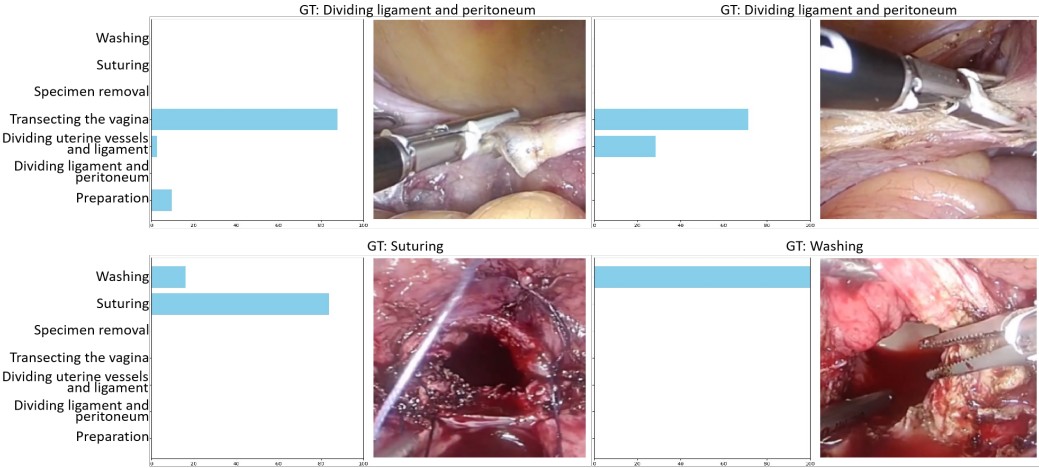

Figure 5: Qualitative surgical phase recognition results on hysterectomy. The y-axis is the class names. The x-axis is the probability of each class. The bottom right image shows that the pretrained model understands the blood fluid.

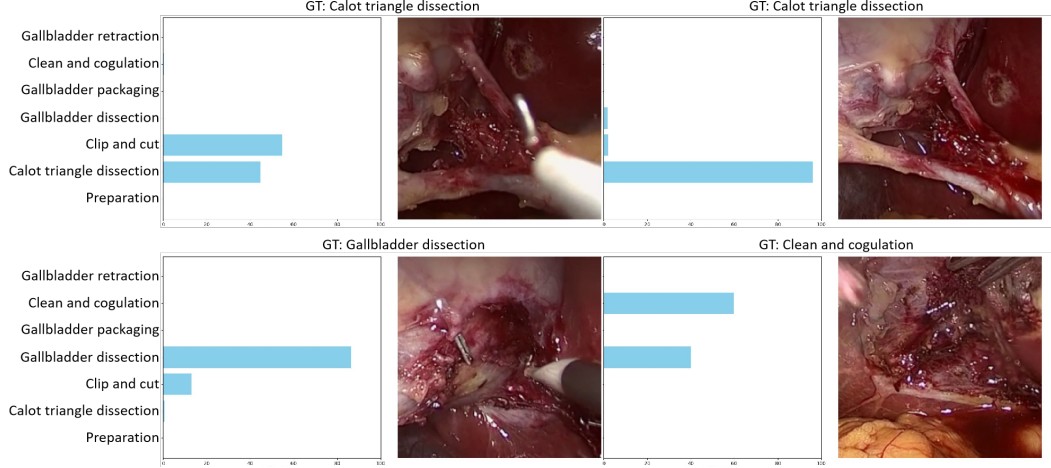

Figure 6: Qualitative surgical phase recognition results on cholecystectomy. The y-axis is the class names. The x-axis is the probability of each class. We find that the pretrained model is triggered by the instrument occurrence, such as hook in the second row.

and abstract texts need to be modified to fit the other video-language pretraining datasets [4, 79] as their hierarchical paired texts are annotated manually. Instead, our knowledge augmentation is more suitable for videos in the wild from online platforms. To address these limitations, future work could focus on developing a general textual augmentation strategy using the LLM's internal knowledge, adapting to the instructional videos that miss keystep and abstract text descriptions. Furthermore, techniques for decentralizing the video-language pretraining could be explored, aiming to pretrain with multi-centric vision-language samples while preserving privacy using the federated learning strategy. This could address the scaling problem in surgical vision-language pretraining and improve the generalizationability across the centers.

## H Knowledge Augmentation

**Build Surgical Knowledge Base.** In Fig. 7, we show that the internal surgical knowledge of large language models can be elicited to build the external knowledge base.

**Build Surgical Knowledge Base.** In Fig. 8, Fig. 9 and Fig. 10, we show that the knowledge of large language model can be used to enrich the semantics of the hierarchical texts, i.e., narrations, keysteps, and abstracts. Notably, it can explain high-level keystep words into descriptive sentences, enhancing textual diversity and preventing overfitting.

**Laparoscopic right colectomy for cecal cancer**

1. Position the patient appropriately for a laparoscopic right colectomy. This usually involves placing the patient in the supine position.
2. Using a trocar, create an access point to the abdomen to allow for the insertion of the laparoscope.
3. Through the laparoscope, verify the presence of the cecal cancer and its location on the right colon.
4. Identify the medial approach and begin the mobilization of the mesentery. Carefully navigate the laparoscopic tool to disconnect the right side of the colon from the rest of the organ.
5. Roam around delicately to identify the key anatomic landmarks. This could include the superior mesenteric vein, ileocolic vessels, or the duodenum.
6. Complete the division of the mesentery intracorporeally. Separate the right colon from the rest of the bowel and carefully preserve the oncologic clearance.
7. Cut the anastomosis stapler to release the healthy section of the right colon.
8. Extract the resected right colon extracorporeally through a small suprapubic incision. Take caution to make as small an incision as possible to ensure minimal harm to the patient.
9. Complete the stapled anastomosis extracorporeally. Connect the healthy section of the colon back to the rest of the organ.
10. After ensuring the anastomosis is secure and not leaking, remove the laparoscope.Please note: This is a broad outline of the steps undertaken during a laparoscopic right colectomy for cecal cancer. The specific steps may vary based on surgeon's expertise, patient's anatomy, and clinical situation.

**Redo Nissen fundoplication with stapled-wedge Collis gastroplasty**

1. Start the procedure by taking down the previous fundoplication.
2. Follow this by identifying the mechanism underlying the failure of the initial repair.
3. Perform an extensive mobilization of the esophagus through the hiatus to achieve an adequate length of intra-abdominal esophagus.
4. Despite the mobilization, if the esophagus remains too short, perform a Collis gastroplasty using the wedge gastrectomy technique over a 50 French bougie.
5. Following the gastroplasty, a 2.5cm of tension-free intra-abdominal esophagus should be achieved.
6. Repair the hiatus with interrupted non-absorbable sutures.
7. Finally, perform a standard Nissen fundoplication.

**Stepwise approach for laparoscopic reversal of Hartmann's procedure**

1. Position the patient on the operating table after administering general anesthesia to ensure patient comfort and positioning.
2. Establish pneumoperitoneum via a Veress needle to inflate the abdomen, creating a space in which to work.
3. Insert three trocars (ports) into the patient's abdomen to allow for the passage of laparoscopic instruments.
4. Inspect the abdomen with a laparoscope to locate the previous colonic stump and assess adhesions and general abdominal conditions.
5. Begin the process of adhesiolysis, involving the careful separation of adhesions between the abdominal wall and the colon.
6. Proceed with the mobilization of the colon by carefully performing a medial-to-lateral dissection.
7. Divide the colon intra-abdominally using a laparoscopic stapler, which seals off the colon and prevents leakage of bowel contents.
8. Identify the rectal stump and mobilize it within the pelvis in readiness for the reconnection of the bowel.
9. An anastomosis (connection) is created between the divided colon and the rectal stump, restoring intestinal continuity.
10. Secure the anastomosis by placing sutures and applying surgical staples to ensure a secure connection with no leakage.
11. Inspect the whole abdominal cavity visually with the laparoscope checking for any signs of bleeding, injury or any overlooked issue before ending the procedure.
12. The trocars are then removed, and the incisions sutured. The pneumoperitoneum is deflated.
13. Clean the surgical area thoroughly.
14. Dress the post-operative wounds correctly.

**Laparoscopic extraction of a CBD stone after failure of ERCP (duodenal perforation)**

1. The surgical area is prepared and patient is positioned for laparoscopic common bile duct (CBD) exploration.
2. Trocars are inserted at suitable locations in the abdominal region to carry out the procedure.
3. The gall bladder is reached and exposed utilizing laparoscopic tools.
4. The cystic duct is identified through careful maneuvering with laparoscopic instruments.
5. A trans-cystic approach is taken to explore the Common Bile Duct.
6. In case of large bile duct stones which cannot be extracted through the cystic duct, a choledochotomy is performed.
7. The CBD stone is visually located using the laparoscopic camera.
8. Laparoscopic instruments are used to extract the stone from the Common Bile Duct.
9. The stone is securely extracted from the body through the previously created trocar incisions.
10. Once the stone is completely removed, the common bile duct and cystic duct are checked for any potential remaining stones or blockages.
11. Procedure concludes with the removal of all laparoscopic tools and the closure of all incisions.

Figure 7: Example of surgical step knowledge base based on the large language models.

1. **Source:** and this be for the so be I cut the mesh just in the middle about seven centimeter link
2. **Target:** Select a mesh of appropriate dimensions that completely covers the hernia defect and extends at least 3 centimetres beyond the defect in all directions
3. **Source:** inferior epigastric vessel come from here
4. **Target:** Utilize dissection instruments to make an opening between the preperitoneal space and the transversalis fascia for easy access to the inguinal region
5. **Source:** the plain zero be often very thickened in this inflammatory condition and capsule dissection must be perform in order to help we find the plain and continued dissection
6. **Target:** Utilize meticulous dissection techniques to divide the blood vessels close to the bowel, ensuring minimal damage to the surrounding area
7. **Source:** the sigmoid colon be now or most completely release from the lateral side wall
8. **Target:** Identify and diagnose the patient with diverticulosis and chronic colo-vesical fistula
9. **Source:** we can morgue correctly define the way to proceed with the dissection
10. **Target:** Proceed with a combination of lateral and medial approach for the mobilization of the mesocolon
11. **Source:** a percutaneous suture use a straight needle be insert in the epigastric region and pass towards the apex of the right carotid
12. **Target:** Begin by positioning three ports (5mm, 12mm, 5mm) in the abdomen for laparoscopy
13. **Source:** middle colic vessel be clip and divide just above the body of the pancreas
14. **Target:** Locate the line of demarcation for the resection, ensuring to capture all the polyps and the other lesion sites observed during the preoperative investigations

Figure 8: Knowledge augmentation on the narration texts.

1. **Source:** Opening of lesser omentum
2. **Target:** The lesser omentum, a fatty apron-like structure that covers the stomach and first part of the duodenum, is opened to allow access to the stomach
3. **Source:** Start of gastric tubulization
4. **Target:** At this step, the surgeon begins creating a tube-like shape from the remaining portion of the stomach, also known as gastric tubulization
5. **Source:** End of tubulization
6. **Target:** This is when the surgeon completes the tubulization process, finalizing the smaller, sleeve-like shape of the stomach
7. **Source:** Division of greater omentum
8. **Target:** In this step, the surgeon divides the greater omentum, a large apron-like fold of visceral peritoneum that hangs down from the stomach
9. **Source:** Jejunojejunostomy
10. **Target:** The surgeon creates an opening in the two loops with a cautery hook for passage of the linear stapler and closes the opening using absorbable sutures
11. **Source:** Gastrojejunostomy
12. **Target:** The surgeon executes the gastrojejunostomy using a circular stapler, creating a connection between the stomach and jejunum
13. **Source:** Closure of Petersen's defect
14. **Target:** Towards the end, the surgeon closes Petersen's space, a potential space after Roux-en-Y gastric bypass, to prevent internal herniation
15. **Source:** Anvil placement
16. **Target:** The end of a nasogastric tube, attached to the anvil, is passed down from the mouth into the stomach
17. **Source:** Division of the ileocolic vessels
18. **Target:** The surgeon separates the blood vessels connected to the ileum and colon to prevent bleeding during the procedure
19. **Source:** Preparing the anastomosis
20. **Target:** The surgeon prepares for the anastomosis, or the surgical connection between two parts of the intestine

Figure 9: Knowledge augmentation on the keystep texts.

1. **Source:** This edit of a live operation demonstrates the performance of a laparoscopic gastric bypass. It demonstrates nicely manoeuvres such as retrocolic placement of the Roux limb and hand-sewn gastrojejunal anastomosis
2. **Target:** This video shows a laparoscopic gastric bypass surgery, focusing on stomach and duodenum procedures and bariatric surgery techniques for morbid obesity treatment. Main activities involve the retrocolic placement of the Roux limb and hand-sewn gastrojejunal anastomosis. They demonstrate the techniques and maneuvers used during this surgery
3. **Source:** This video shows the case of a female patient presenting with a low rectal cancer for which neoadjuvant therapy is used. The author performs a totally laparoscopic TME using a medial approach. A colorectal anastomosis without bowel protection is performed
4. **Target:** This is a surgical lecture video on a laparoscopic low anterior resection with Total Mesorectal Excision (TME) and medial mobilization of the splenic flexure in a female patient. This procedure is utilized to treat a low rectal cancer and involves the use of a medial approach. The video details how to perform a colorectal anastomosis without bowel protection. The procedure is entirely laparoscopic
5. **Source:** In this live educational video, Professor Himpens presents the case of a 34-year-old female patient (BMI of 41) with a history of morbid obesity since adolescence. She will undergo a laparoscopic sleeve gastrectomy (LSG). The preoperative work-up was normal. She had lost 2Kg six months before the procedure. Nowadays, laparoscopic sleeve gastrectomy (LSG) is one of the most commonly performed bariatric procedures. Surgical pitfalls are emphasized during the video to make sure that LSG is achieved adequately and to prevent any potential complications. In addition, trocars placement, location of the first firing of the linear stapler, the reasons why oversewing of the staple line is not performed, and thrombosis prophylaxis are also discussed during the procedure
6. **Target:** This educational video demonstrates a laparoscopic sleeve gastrectomy for a morbidly obese patient. The surgical procedure involves techniques such as the placement of trocars and the first firing of the linear stapler. It also addresses potential surgical pitfalls to ensure the adequate execution of the procedure and prevention of complications. The video highlights that oversewing of the staple line isn't performed during the procedure and also discusses the methods for thrombosis prophylaxis
7. **Source:** Intrathoracic migration of the fundoplication is one of the most common causes of failure after antireflux surgery. When the patient develops symptoms related to the volume of intramediastinal hernia, the only option is to reoperate. Such redos are complex and necessitate a thorough and painstaking approach to the potential underlying mechanisms causing intrathoracic migration, namely the length of the esophagus and cruroplasty
8. **Target:** This surgical video falls under the categories of stomach and duodenum, hiatal hernia, reflux, Nissen fundoplication, and hernia surgery. The video demonstrates a reoperation for symptomatic intrathoracic migration of a fundoplication, involving valve repositioning and reinforced crural repair. The principal activities consist of examining the underlying mechanisms causing intrathoracic migration such as the length of the esophagus and cruroplasty
9. **Source:** This video demonstrates our transumbilical three-trocar technique for single incision total colectomy and partial proctectomy with intracorporeal side-to-end ileorectal anastomosis using standard laparoscopic instrumentation. The patient is a thin 19-year-old boy with a BMI of 19 presenting with familial adenomatous polyposis (FAP). The previous colonoscopy has shown 300 polyps in the colon and very few in the distal rectum. Conventional trocars (5mm, 10mm, and 12mm) are used through a 3.5cm transumbilical incision. The ligation of the vessels is mostly carried out by the Ligasure-V vessel-sealing device using a medial-to-lateral approach. The specimen is extracted through the umbilical incision after removal of the 10mm and 12mm cannulas. The ileorectal anastomosis is carried out intracorporeally using a double stapling technique
10. **Target:** The video shows a transumbilical single incision laparoscopic total colectomy and partial proctectomy with ileorectal anastomosis performed on a 19-year-old patient with familial adenomatous polyposis. The surgery primarily uses a three-trocar technique and standard laparoscopic instruments including Ligasure-V vessel-sealing device for ligating vessels. The surgery involves making a 3.5cm transumbilical incision using 5mm, 10mm, and 12mm trocars. The colectomy specimen is extracted through the same umbilical incision. The final ileorectal anastomosis is achieved intracorporeally employing a double stapling method

Figure 10: Knowledge augmentation on the abstract texts.

