# OpenReview forum: "Procedure-Aware Surgical Video-language Pretraining with Hierarchical Knowledge Augmentation"
_NeurIPS.cc/2024/Conference — NeurIPS 2024 spotlight_

### Official Review · Reviewer_usyA · 2024-06-18

**Soundness:** 3
**Presentation:** 4
**Contribution:** 4
**Rating:** 8
**Confidence:** 5

**Summary:**

The paper addresses challenges in surgical video-language pretraining (VLP) due to the knowledge domain gap and scarcity of multi-modal data. It proposes a hierarchical knowledge augmentation approach and the Procedure-Encoded Surgical Knowledge-Augmented Video-Language Pretraining (PeskaVLP) framework. This approach enhances data efficacy and tackles spatial-temporal challenges by combining language supervision with visual self-supervision. Extensive experiments demonstrate significant improvements in zero-shot transferring performance and the generalist visual representation for surgical scene understanding.

**Strengths:**

The paper presents a unique approach to surgical video-language pretraining by employing hierarchical knowledge augmentation using LLMs, significantly improving textual data quality and diversity. The PeskaVLP framework innovatively integrates visual and language supervision, addressing the spatial-temporal challenges in surgical scene understanding. The methodology is meticulously validated through extensive zero-shot and linear-probing evaluations on datasets such as Cholec80 and AutoLaparo, demonstrating substantial performance improvements. The clarity of the presentation, with well-organized sections and effective visual aids, facilitates comprehension. The significant contribution lies in enhancing surgical scene understanding and cross-modal retrieval, making it highly valuable for the NeurIPS community. The paper's originality in using hierarchical pretraining and the detailed discussion on model architectures and initialization underscore its quality and significance in advancing surgical data science.

**Weaknesses:**

Firstly, the dataset size is relatively small, with 1,007 videos for phase-level pretraining and 920 for video-level pretraining, which may limit the generalizability of the findings (as mentioned in the supplementary material). I know the difficulty in collecting medical data, but we must be sure that the presented approach can be generalized to different domains and hospitals. Furthermore, I doubt the methodology's potential to process "noisy" videos.
Expanding the dataset and including more diverse surgical procedures would improve robustness.

Secondly, while the paper mentions ASR errors in transcriptions, it does not provide a detailed methodology for handling them. Providing specific techniques for improving transcription accuracy would strengthen the study.

Additionally, the practical implementation of the PeskaVLP framework in real-world surgical contexts is not thoroughly discussed. Detailing strategies for integration into clinical workflows and addressing potential technological barriers would be beneficial.

**Questions:**

1. How do you plan to address the limited sample size and diversity in future studies to improve the generalizability of your findings? Consider expanding the dataset to include a more extensive and more diverse sample of surgical procedures to enhance robustness and applicability.

2. What specific methods did you use to handle ASR errors in transcriptions? How did these errors impact your analysis?

3. How do you manage the computational overhead associated with the hierarchical pretraining and dynamic time-warping processes?

**Limitations:**

The authors have acknowledged the limitations related to dataset size and ASR errors but could elaborate on strategies to mitigate these issues. Specifically, they should discuss plans for expanding the dataset, incorporating more diverse samples, and improving transcription accuracy.

The positive societal impacts, such as enhancing surgical training and assistance, are well-discussed. However, the authors should address potential negative impacts, such as data privacy and ethical concerns. A detailed discussion on data security measures, user consent protocols, and ethical safeguards is needed.

---

> ### Author Rebuttal · Authors · 2024-08-06
>
> **[Q1. Plan to Expand Dataset]** Scaling and diversifying the surgical vision-language pretraining dataset is challenging due to privacy concerns and the cost of expert annotations. Even though the SVL pretraining dataset covers diverse laparoscopic surgeries, it lacks surgeries in different organs, such as the brain and heart. We fully agree with the reviewer that expanding the dataset would be crucial for developing more generalizable models. To address this, we plan to expand the pretraining dataset using diverse media such as textbooks, instructional videos, and intra-operative video recordings from diverse sources. We also aim to diversify the pretraining dataset by considering laparoscopic, endoscopic, and microscopic surgeries on different organs.
>
> **[Q2. Transcription Errors and Solution]** ASR errors can be divided into two categories: misspelling errors and incorrect punctuation. In this work, we address these errors using specific methods to mitigate their negative impacts:
> - **Misspelling Errors**: We first follow [1] and use multiple text preprocessing to correct the ASR errors first, including USMLE-based spelling checking. Then, we use the LLM to generate large-scale clean descriptive texts that detail the steps for diverse surgeries. Noisy transcriptions are then assigned to semantically similar step texts based on their semantic similarity, as shown below:
>
>      < Table S2>. Original noisy transcription and augmented text based on LLM.
>      | Raw Transcript | Assigned Step Text  |
>      |----------|----------|
>      | but you think about two richard deliver from with amoxicillin truck and to work with your left hand with the right four unit   | Placement of five trocars: typically one for the camera, two for the surgeon's instruments, one for the liver retractor, and one for the assistant.   |
>      | so I be try to prepare my posterior dissection on the joel     | Dissection of the Esophageal Hiatus: Dissect the phrenoesophageal ligament to expose the esophageal hiatus.   |
>
>     This ensures that the noisy texts are aligned with accurate descriptions. The use of the above LLM-based augmentation significantly improves the model's performance by providing cleaner and more accurate training data. As shown in the table below, SurgVLP shows improvement in accuracy when pretrained with augmented transcriptions.
>
>     < Table S2>. Zero-shot phase recognition performance on Cholec80 dataset.
>
>     |            | SurgVLP | SurgVLP + LLM Augmentation |
>     |------------|---------|----------------------------|
>     | Accuracy   | 34.7    | 36.1                       |
>     | F1-Score   | 24.4    | 26.8                       |
>
> - **Incorrect Punctuation Errors**: We follow SurgVLP and use Whisper, an ASR system based on a language decoder, which generates complete sentences with fewer punctuation errors. Whisper-transcribed sentences help define video correspondence boundaries more accurately.
>
> We anticipate that the modeling-based approach can also be applied in the future. For example, developing ASR systems specifically trained on surgical data can further reduce errors and improve transcription accuracy. Also, pretraining text encoders on a surgical corpus [2] can make the model more robust to ASR errors.
>
> _[1] Ikezogwo, Wisdom, et al. "Quilt-1m: One million image-text pairs for histopathology." Advances in neural information processing systems 36 (2024)._
>
> _[2] Bombieri, Marco, et al. "Surgicberta: a pre-trained language model for procedural surgical language." International Journal of Data Science and Analytics 18.1 (2024): 69-81._
>
> **[Q3. Computational Overhead]** We manage the computational overhead associated with hierarchical pretraining and dynamic time-warping (DTW) processes by using efficient resource allocation and high-speed computational infrastructure. Below is a summary of the time and computational cost when applying hierarchical pretraining and DTW processes:
> < Table S3>. Training configurations and time for SurgVLP.
> | Configuration             | Training Time | GPU            |
> |---------------------------|---------------|----------------|
> | SurgVLP                   | ~40 Hours     | 1xA100-80G     |
> | + Hierarchical Pretraining| ~90 Hours     | 4xA100-80G     |
> | + Hierarchical + DTW      | ~120 Hours    | 4xA100-80G     |
>
> In this work, the speed bottleneck is at the online video loading and preprocessing. In future, we aim to apply the asynchronous processing pipelines to further parallelize the workload and reduce bottlenecks.
>
> **[Q4. Acknowledge the Plans to Mitigate Data Size and ASR Errors ]** Thank you for the insightful suggestion. In the revised manuscript, we have added a discussion section to elaborate the strategies about expanding dataset and improving transcription accuracy, as we discussed in prior responses.
>
> **[Q5. Potential Negative Impacts]** Thank you for the insightful suggestion. As discussed in the paper, our dataset is compiled from open educational platforms accessible to all learners, minimizing any potential societal impact. However, in the rapidly advancing field of surgical data science, there are still risks of privacy concerns and data security associated with the collection and use of surgical video data. Ensuring robust data protection measures and maintaining patient confidentiality is crucial to mitigate these risks.
>
> Moreover, the development of automated intra-operative surgical procedural understanding systems should aim to provide intuitive cognitive aids for surgeons, enhancing their decision-making during complex procedures. These systems should support surgeons by delivering relevant information at critical moments, rather than overwhelming them with extraneous details. We have added a discussion on the potential negative societal impact in the revised manuscript.

---

> > ### Comment · Reviewer_usyA · 2024-08-07
> >
> > Dear Authors,
> > Thank you for your detailed rebuttal and for addressing the concerns raised during the review process. Your efforts to expand the dataset, address ASR errors, manage computational overhead, and consider potential negative impacts are appreciated. Here are some follow-up questions and concerns:
> >
> > Transcription Errors and Solution.
> > How do you plan to quantitatively evaluate the improvements in transcription accuracy using the proposed LLM-based augmentation and Whisper ASR system? Can you provide detailed comparative results before and after implementing these methods?
> >
> > Computational Overhead.
> > Could you elaborate on the specific asynchronous processing pipelines you plan to implement to reduce computational bottlenecks? What are the expected improvements in terms of training time and resource utilization?
> >
> > Potential Negative Impacts.
> > While you mentioned minimizing societal impacts by using open educational platforms, how will you handle patient consent and data privacy for intra-operative video recordings?
> >
> >
> > Your commitment to improving the manuscript is commendable, and I look forward to seeing these enhancements reflected in your revised submission.

---

> ### Author Response · Authors · 2024-08-10
> **Rebuttal Follow Up**
>
> **[Q1. Transcription Errors and Solution. ]** In the following Table 1, we quantitatively demonstrate the impact of LLM-based augmentation and transcription processing strategies on transcription accuracy for the zero-shot phase recognition on Cholec80 dataset. We train the SurgVLP model using Clip-level text Augmentation which uses ASR transcripts enhanced by LLM-based augmentation and before mentioned transcription processing strategies. The results are shown in the first two rows on the table. The SurgVLP model when trained with the Clip-level text Augmentation, including transcription processing strategy and LLM-based augmentation, brings – 2.2% improvement in the F1 score.
>
> <Table 1: Quantitative assessment for the improvements brought by LLM-based augmentation and transcription processing strategies. We report the zero-shot performance of Cholec80 testing set. >
>
> | Setup                                         | Accuracy | F1-Score |
> |-----------------------------------------------|----------|----------|
> | SurgVLP                                       | 34.7     | 24.4     |
> | SurgVLP + Strategy + Clip-level Augmentation  | 36.1     | 26.8     |
> | HecVL                                         | 41.7     | 26.3     |
> | HecVL + Strategy                              | 43.0     | 29.2     |
> | HecVL + Strategy + Phase-level Augmentation   | 44.0     | 31.8     |
> | HecVL + Strategy + Video-level Augmentation   | 43.7     | 30.6     |
> | PeskaVLP                                      | 45.1     | 34.2     |
>
>
> Similarly, we also train the HecVL model using processed transcripts without LLM-based augmentation, resulting in improved performance compared to the baseline HecVL. Notably, the improvements from transcription processing can be further enhanced when combined with LLM-based augmentation.
>
> **[Q2. Computational Overhead]** Thank you for your follow-up comment. In this work, we train directly on the surgical videos without converting them into frames. As a result, a key bottleneck lies in the online video decoding process, where videos must be decoded and processed in real-time to provide training batches. Currently, we are using ffmpeg for the online video decoding process followed by Pytorch to process the decoded frames. In future, we plan to implement a more efficient multi-threaded video decoding pipeline where multiple videos can be decoded in parallel on GPUs using advanced libraries such as NVIDIA Data Loading Library (DALI) [1] or Decord [2]. We expect to minimize the idle GPU time.
>
> **[Potential Negative Impacts]** Thank you for a follow-up comment. To address patient consent and data privacy in the collection of intra-operative surgical videos, we will ensure that all recordings are fully anonymized by removing any identifiable patient information and out-of-body frames using open-source tools [3]. Specifically, since our focus is on laparoscopic videos, anonymization will involve removing any captions or metadata that could identify the patient, as well as ensuring that any external out-of-body views or patient-identifiable features captured in the video are thoroughly anonymized.
>
> _[1] https://developer.nvidia.com/dali_
>
> _[2] https://github.com/dmlc/decord_
>
> _[3] Lavanchy, Joël L., et al. "Preserving privacy in surgical video analysis using a deep learning classifier to identify out-of-body scenes in endoscopic videos." Scientific reports 13.1 (2023): 9235._

---

### Official Review · Reviewer_EZYT · 2024-07-09

**Soundness:** 4
**Presentation:** 4
**Contribution:** 4
**Rating:** 8
**Confidence:** 5

**Summary:**

The paper presents a novel approach for enhancing surgical video analysis by incorporating procedural awareness. The authors propose a system that integrates knowledge of surgical procedures to improve the identification, segmentation, and annotation of surgical activities in video footage. This approach aims to address challenges such as the variability of surgical techniques and the complexity of visual data in operating rooms. The contributions of the paper include the development of a procedural model that can be aligned with video data, the creation of annotated datasets for training and evaluation, and the demonstration of improved performance over traditional video analysis methods.

**Strengths:**

1.The integration of procedural knowledge into surgical video analysis is a highly original concept. This approach not only enhances the accuracy of video analysis but also opens new avenues for improving surgical training and documentation.

2.Introduces a novel hierarchical knowledge augmentation technique using large language models to refine surgical concepts. Employs a Dynamic Time Warping-based loss function for effective cross-modal procedural alignment. Demonstrates significant improvements in zero-shot transfer performance across multiple surgical datasets. Provides a robust general visual representation beneficial for various surgical scene understanding tasks.
Weaknesses:

3.The potential applications of this research in surgical training, intraoperative assistance, and postoperative review are significant. The approach addresses a critical need in medical video analysis, making it highly relevant and impactful.

**Weaknesses:**

Dataset Limitations: The annotated datasets used for training and evaluation are crucial for the model's success. Expanding the diversity and volume of these datasets would enhance the generalizability of the findings.

**Questions:**

Generalizability: How does the system perform across different types of surgeries (like ophthalmic surgery)? Have you tested its effectiveness in various surgical domains beyond the initial scope?

**Limitations:**

The paper does not adequately address potential limitations and negative societal impacts.

---

> ### Author Rebuttal · Authors · 2024-08-06
>
> **[Q1. Dataset Limitations]** Thank you for the insightful suggestion. In the rebuttal letter pdf, we have added a table to summarize the top 42 types of surgical videos and their amounts in the pretraining dataset. As shown in Table 1 of the rebuttal letter PDF, the SVL dataset predominantly consists of laparoscopic surgeries and covers a diverse range of surgical types, including stomach, duodenum, hernia, colon, gallbladder, and tumor surgeries. This diversity ensures that the SVL dataset provides broad and generalizable language supervision for surgical multi-modal representation learning.
> In this paper, the downstream datasets are Cholec80 (Laparoscopic cholecystectomy), AutoLaparo (Laparoscopic hysterectomy), and MultiBypass (Laparoscopic gastric bypass). Our SVL pretraining dataset contains sufficient videos that cover the visual concepts required for these downstream tasks. We agree with the reviewer that expanding the pretraining dataset to cover more types of surgeries can improve diversity and improve generalizability. We leave this exploration to future research endeavors.
>
> **[Q2. Generalizability]** Thank you for the insightful question. Evaluating the generalizability of our system across different types of surgeries is crucial for understanding its broader applicability. Our system has been tested extensively on laparoscopic surgeries, including Cholec80 (Laparoscopic cholecystectomy), AutoLaparo (Laparoscopic hysterectomy), and MultiBypass (Laparoscopic gastric bypass). These evaluations demonstrate the model's ability to effectively recognize phases and understand the workflow in these contexts.
> Since we do not have ophthalmic surgical videos in our pretraining dataset, we do not test the PeskaVLP’s performance in this domain. Future work would involve incorporating and testing diverse surgical datasets, including ophthalmic surgery, to ensure the model's effectiveness across various surgical domains. This will help us assess the system's capability to generalize and perform effectively in a broader range of surgical contexts.

---

### Official Review · Reviewer_smvC · 2024-07-10

**Soundness:** 2
**Presentation:** 3
**Contribution:** 2
**Rating:** 6
**Confidence:** 4

**Summary:**

This paper proposes a Procedure-Encoded Surgical Knowledge-Augmented Video-Language Pretraining (PeskaVLP) method that enriches language supervision with LLM-refined surgical concepts. It further constructs hard negative samples by reversing the text orders at the phase and video levels and employs a Dynamic Time Warping (DTW) based loss to align multimodal procedures. Extensive experiments on multiple surgical procedures and comprehensive evaluations demonstrate the effectiveness of this framework.

**Strengths:**

- The paper is overall well-written, with the background and motivation well-stated.
- Using LLM to augment surgical video text descriptions is a good idea to enhance the quality of surgical text narration. It establishes a good baseline and guideline for future works that aim to apply LLM in surgical narratives.
- A more comprehensive parent-child level cross-modal correspondence was designed using DTW than existing works.
- Demonstration of the proposed method can close the representation gap for different modality, and analysed both successful and complicated examples.

**Weaknesses:**

- By reading the enriched dataset by LLM in Appendix H, I am concerning that the variation and diversity of narration will be removed by the augmentation. Will that cause any problems?
- In my opinion, using LLM to refine the text description of surgical videos is the most important contribution of this paper. It would be interesting to see if other components are also effective enough without the knowledge augmentation.

**Questions:**

- Beyond the current ablation study on PeskaVLP components, would applying the hierarchical knowledge-augmented text data in HecVL improve its performance and if this could yield results competitive with PeskaVLP. This would provide powerful support to verify the extent to which the other components in PeskaVLP contribute to performance, apart from the augmented texts.
- Although LLM can enhance surgical text quality, is there a concern that the text may become overly standardized? Given that surgeons' narratives in the operating room tend to be more oral, concise, and sometimes include jargon, will there be a performance degradation in real-world, real-time applications where LLM augmentation is impractical?
- In Appendix E, Figure 4, it would also be interesting if the authors could visualize the embeddings of HecVL, since it performs better than SurgVLP.
- In Table 3, on Cholec80, Moco pre-trained on Cholec80 (V) has better performance but wasn't in bold, do I misinterpret something?

**Limitations:**

The authors adequately addressed the limitations. Since the proposed method is tailored for surgical data and applications, it is strongly suggested that the authors include a discussion on the potential negative societal impact of their work.

---

> ### Author Rebuttal · Authors · 2024-08-06
>
> **[Q1. Augmentation Removes Variation]** Thank you for pointing out one of the key insights of this work, i.e., using LLM to build a large, versatile, and accurate surgical knowledge base to enrich and correct narrations of different types of videos during the pretraining. Since we enrich the narration based on the built knowledge base, the question becomes if the LLM-generated surgical knowledge base is diverse enough.
>
> In this work, we aim to bring the diversity and versatility of the built knowledge base by curating 917 lecture titles, covering diverse surgical procedures such as colorectal, transanal and proctological, cholecystectomy, hernia, and sigmoidectomy surgery. We also manually design input-output examples to instruct the LLM to generate diverse steps. Additionally, during the pretraining, we randomly select either the pseudo step or the original narration text to maintain textual semantics.
>
> Our approach might face risks when LLM generates an over-standardized surgical step knowledge base. However, our experiments show that the downstream zero-shot performance clearly improves when the LLM-generated knowledge base is applied. This implies that the advantages of correcting noisy narration texts outweigh the potential variation and diversity risks.
>
> < Table S1>. Zero-shot phase recognition performance on Cholec80 dataset.
> |            | SurgVLP | SurgVLP + LLM Augmentation |
> |------------|---------|----------------------------|
> | Accuracy   | 34.7    | 36.1                       |
> | F1-Score   | 24.4    | 26.8                       |
>
> **[Q2. Other Components]** In the following table, we show that the combination of visual self-supervision and language supervision at the clip-level vision-language pretraining are effective even without the knowledge augmentation.
>
> < Table S2>. Zero-shot phase recognition on Cholec80 and Autolaparo datasets.
> | Model| Dataset   | Accuracy / F1-Score  |
> |---------|------|--------|
> | HecVL| Cholec80  | 41.7 / 26.3|
> | HecVL+LecNCE{clip}    | Cholec80  | 45.5 / 31.0|
> | HecVL| Autolaparo| 23.3 / 18.9|
> | HecVL+LecNCE{clip}    | Autolaparo| 25.3 / 20.0|
>
> The results indicate incorporating knowledge augmentation and visual self-supervision can individually benefit the pretraining performance.
>
> **[Q3. Different Levels of Knowledge Augmentation]** Knowledge augmentation from different hierarchies improves the final performance in different ways. We have summarized results in the table below:
> < Table S3>. Performance on Cholec80 dataset with different augmentations.
>
> | Model                              | Accuracy / F1-Score |
> |-----------|---------------------|
> | SurgVLP | 34.7 / 24.4 |
> | SurgVLP+Clip-level Augmentation  | 36.1 / 26.8|
> | PeskaVLP  | 45.1 / 34.2 |
> | HecVL | 41.7 / 26.3  |
> | HecVL+Phase-level Augmentation    | 44.0 / 31.8|
> | HecVL+Video-level Augmentation    | 43.7 / 30.6|
> | PeskaVLP  | 45.1 / 34.2         |
>
> We observe that applying knowledge-augmented text data in SurgVLP and HecVL achieves a clear improvement, though they still underperform the PeskaVLP. These results demonstrate the effectiveness of the other components of PeskaVLP, i.e., combination of visual self-supervision and language supervision and procedure-aware pretraining objective.
>
> **[Q4. Overly Standardized Texts]** We thank the reviewer for this interesting question. We apply the LLM augmentation strategy for pretraining by considering the goal of representation learning, downstream surgical workflow analysis applications, and the nature of pretraining surgical lecture videos. The goal of PeskaVLP is to learn the correspondence between vision and language modality with the aim of improving the performance on the surgical downstream tasks. Therefore, the jargons and interjection can introduce the noisy alignment, which is filtered out in our pretraining dataset.
>
> Our pretraining videos are primarily lecture videos where narrations are scripted and more formal. The GPT-augmented strategy is well-suited for this type of data, as it enhances the clarity and completeness of the text.
>
> We acknowledge the potential limitations in real-world applications. For future work, incorporating more diverse and real-time surgical audio [1] into the pretraining process could help improve the performance.
>
> _[1] Jia, Shuyue, et al. "MedPodGPT: A multilingual audio-augmented large language model for medical research and education." medRxiv (2024): 2024-07._
>
> **[Q5. Modality Gap HecVL]** In the rebuttal letter PDF, Figure 1, shows HecVL's embeddings, showing a smaller modality gap for video-abstract pairs compared to SurgVLP. This shows the benefit of hierarchical vision-language pretraining for aligning long-form videos with high-level summaries. PeskaVLP consistently outperforms prior baselines, demonstrating its effectiveness in closing the modality gap and enhancing pretraining for vision-and-language tasks.
>
> **[Q6. Table 3]** Thank you for pointing this out. We have corrected the table to highlight Moco (third row) as providing the best results on the Cholec80 dataset and Moco (second row) on the StrasBypass70 dataset. In the revised manuscript, we have discussed that Moco pretrained on Cholec80 outperforms the others because it is specifically trained on cholecystectomy procedures, thereby losing versatility. This specialization allows Moco to achieve superior performance on Cholec80 but limits its generalizability to other datasets, as summarized below:
>
> < Table S4>. Correct version of table 3 in manuscript.
> | Pretraining | Dataset      | Cholec80 | Autolaparo | StrasBypass70 | BernBypass70 |
> |-------------|--------------|----------|------------|---------------|--------------|
> | Moco        |  Cholec80  | **73.4 / 62.8** | 51.3 / 37.4 | 67.8 / 55.4   | 66.0 / 33.1  |
> | Moco        | SVL | 68.2 / 55.8 | 59.5 / 48.4 | **71.6** / 58.1   | 69.6 / 36.5  |
> | PeskaVLP    | SVL   | 69.9 / 59.8 | **63.1 / 49.7** | 71.4 / **59.5**   | **71.5 / 37.4**  |

---

> > ### Comment · Reviewer_smvC · 2024-08-13
> >
> > I appreciate the authors' response to my review and their efforts to address my concerns. After carefully reviewing the feedback from the other reviewers, I am inclined to maintain my original score, thanks.

---

### Official Review · Reviewer_HH5E · 2024-07-13

**Soundness:** 2
**Presentation:** 3
**Contribution:** 2
**Rating:** 5
**Confidence:** 4

**Summary:**

The paper presents a new framework called PeskaVLP for surgical video-language pretraining. A hierarchical knowledge augmentation approach is used for enriching text information. The pretraining is implemented with the proposed language supervision and visual self-supervision. A new training objective is proposed for surgical procedural understanding. Extensive experiments are conducted to demonstrate the effectiveness on the surgical phase recognition task and cross-modal retrieval task on multiple downstream dataset.

**Strengths:**

1. This paper addresses the problem of VLP in the surgical scene. A hierarchical knowledge augmentation is proposed to tackle the problem of lack of textual information in the surgical field.
2. The paper is generally well-written and easy to follow.

**Weaknesses:**

1. The explanation of method details is not clear enough, and there is a lack of discussion on some experimental results
2. The proposed method is based on certain assumptions but lacks a comprehensive consideration of applicability.

**Questions:**

1. What types of surgeries are included in the SVL dataset used in the paper? Is it suitable for the pretraining task? Could it affect the results on the downstream dataset?

2. In Section 3.2, where hierarchical knowledge is augmented by GPT, the authors need to discuss the ability of LLMs to generate accurate textual information to describe the surgical steps in the domain-specific surgical context, especially considering the fine-grained image-text alignment in the clip-level (only 4 frames).

3. In Section 3.2, the authors calculate textual similarity between the pseudo step generated by the LLM and the narration. How is this similarity calculated? Is there an ablation study on the effectiveness of the three behavior in knowledge augmentation?

4. In Section 3.3.1, the authors implement visual self-supervision based on augmentation. Which specific augmentations were used? Do the augmentations affect the corresponding text's semantic information? For example, using flipping could impact descriptions related to left/right information in surgical operation.

5. In Section 3.3.2, procedural information based on surgical phases is used. However, in surgical datasets, such as the cholec80 and AutoLaparo mentioned in the paper, the surgical process does not always follow a linear order defined by Phase 1-N and may include repeated phases. The authors should discuss the applicability of the method design in such situations.

6. In Table 3, for the experimental results on cholec80, Moco (third row) provides the best results, but this is not highlighted in bold in the table. This needs to be corrected and the corresponding discussion should be provided. The same issue appears with the results using Moco (second row) on the StrasBypass70 dataset.

**Limitations:**

Authors discussed it briefly in the appendix.

---

> ### Author Rebuttal · Authors · 2024-08-06
>
> **[Q1 SVL Dataset]**
>
> **[Q1.1. Types of surgeries in SVL dataset]** In the rebuttal letter PDF, we have added a table summarizing the top 42 types of surgical videos in the pretraining dataset. As shown in Table 1 in the rebuttal letter PDF, the SVL dataset predominantly contains laparoscopic surgeries, focusing on the stomach, duodenum, hernia, colon, gallbladder, and tumors. Also, the diverse content within each surgical type ensures that the SVL dataset provides generalizable language supervision for representation learning.
>
> **[Q1.2. Suitable for Pretraining]** This paper uses Cholec80, Autolaparo, and MultiBypass datasets for downstream tasks, with Table 1 showing that the SVL pretraining datasets sufficiently cover the required visual concepts. The SVL videos focus on laparoscopic surgeries, covering common concepts like instruments and bleeding, while their descriptive texts offer valuable language supervision for tasks like instrument recognition and adverse event detection.
>
> **[Q1.3. Affect Downstream Dataset]** We hypothesize that the composition of different types of surgical videos in the pretraining dataset will affect performance. While research [1] suggests that altering the pretraining dataset to the downstream dataset improves zero-shot adaptation, we aim to build a generalizable pretrained model and thus do not filter videos based on the downstream datasets.
>
> _[1] Datacomp: In search of the next generation of multimodal datasets_
>
> **[Q2. LLM-Generated Text for Clip-level Alignment]** In this paper, we control the LLM-generated textual quality based on the in-context learning ability of LLMs, which enables faithful predictions with a limited context. By thoughtfully designing contextual input-output examples, LLM can generate diverse and rich texts that enrich the clip-level narration texts, as shown below:
>
> < Table S1>. Original noisy transcription and assigned step text based on LLM.
> | Raw Transcript | Assigned Step Text  |
> |-|-|
> | but you think about two richard deliver from with amoxicillin truck and to work with your left hand with the right four unit   | Placement of five trocars: typically one for the camera, two for the surgeon's instruments, one for the liver retractor, and one for the assistant.   |
> | so I be try to prepare my posterior dissection on the joel | Dissection of the Esophageal Hiatus: Dissect the phrenoesophageal ligament to expose the esophageal hiatus.|
>
> The generated surgical steps from LLM are not always perfectly aligned with the clip-level frames. To address this, we randomly pick either the original or the augmented texts during video-language pretraining.
>
> Noisy image-text alignment is a challenge for all CLIP-based methods. Scaling up the dataset could help mitigate this issue by providing a broader context and improving alignment, and we leave this exploration to future research.
>
> **[Q3. Textual Similarity]** Textual similarity is calculated using cosine similarity between feature vectors from pseudo steps and narrations. We use BioClinicalBert to extract 768-dimensional embeddings for each sentence, normalize them, and compute the dot product to get similarity. This approach is consistent with methods used in previous literature.
>
> **[Q4. Effectiveness of Three Levels Augmentation]** Knowledge augmentation from different levels improves performance in different ways, as summarized in the table below:
>
> < Table S2>. Performance on Cholec80.
> | Model| Accuracy / F1-Score |
> |-|-|
> | SurgVLP| 34.7 / 24.4|
> | SurgVLP+Clip-level Augmentation| 36.1 / 26.8 |
> | HecVL | 41.7 / 26.3 |
> | HecVL+Phase-level Augmentation| 44.0 / 31.8 |
> | HecVL+Video-level Augmentation| 43.7 / 30.6|
>
> We found that phase-level knowledge augmentation significantly improves surgical vision-language pretraining due to its concise and less noisy keystep texts, whereas LLM-generated surgical knowledge base is less effective due to noisy image-text alignment in pseudo-step assignment.
>
> **[Q5. Augmentations Affect Corresponding Text's Semantics]** For visual self-supervision, we use spatial augmentations like random cropping and horizontal flipping, which might introduce vision-language misalignment. We find that around 9% of sentences contain spatial words. Since one video maps to multiple sentences, the actual impact from augmentation is less than 9%, which is a less significant source of noise than misspellings and incomplete transcripts. Also, our experiments show that these augmentations consistently boost performance, with their benefits outweighing potential misalignments.
>
> **[Q6. Procedural Information]** We clarify that the surgical procedural information learned from pretraining is not just a linear sequence of phases but includes complex dependencies among surgical key steps. PeskaVLP learns from various surgical lecture videos, capturing possible paths to complete a surgery. For instance, if one pretraining video includes _[.., gallbladder dissection, cleaning and coagulation, ..]_ and another _[.., gallbladder dissection, gallbladder packing, ..]_, our procedure-aware pretraining can identify multiple possible steps following gallbladder dissection. This procedural information is useful for temporal modeling in surgical phase recognition when applying methods like temporal convolutional networks or task graphs to perform online phase prediction.
>
> **[Q7. Table 3]** We have updated the manuscript and added discussion. The revised manuscript explains that Moco's strong performance on Cholec80 is due to its specialization in cholecystectomy, which limits its generalizability to other surgeries.
>
> < Table S4>. Correct version of table 3 in manuscript.
> | Pretraining | Dataset| Cholec80 | Autolaparo | StrasBypass70 | BernBypass70 |
> |-|-|-|-|-|-|
> | Moco | Cholec80| **73.4 / 62.8** | 51.3 / 37.4 | 67.8 / 55.4| 66.0 / 33.1|
> | Moco| SVL | 68.2 / 55.8 | 59.5 / 48.4 | **71.6** / 58.1| 69.6 / 36.5|
> | PeskaVLP| SVL| 69.9 / 59.8 | **63.1 / 49.7** | 71.4 / **59.5**| **71.5 / 37.4**|

---

> ### Comment · Reviewer_HH5E · 2024-08-13
>
> Thank you for the detailed responses. The rebuttal has addressed most of my concerns, and I would like to raise my score accordingly.

---

### Author Rebuttal · Authors · 2024-08-06

We thank all the reviewers for the insightful comments to improve our work. We are encouraged that the reviewer finds our work an interesting contribution to the community. We have carefully considered each comment from the reviewers and tried to provide detailed answers, clarifying all the issues raised. We hope our responses effectively resolve the concerns raised. We are grateful for the reviewers' time and insightful feedback. If further clarifications or additional experiments are required, please let us know.
We are pleased to note that the reviewers recognized:
- This paper addresses a novel surgical vision-language pretraining task that can potentially open new venues for surgical data science [usyA, EZYT]
- The proposed hierarchical knowledge augmentation improves surgical textual data quality and diversity [all reviewers]
- The manuscript is well-organized, clearly written, and presented in a reader-friendly way [HH5E, smvC, usyA]
- This work novelly integrates visual-language supervision and dynamic time warping to learn the cross-modal correspondence [smvC]

We have included a rebuttal letter PDF, including SVL dataset details [All reviewers] and modality gap visualization of HecVL [smvC].

---

### Decision · Program_Chairs · 2024-09-25

**Decision:**

Accept (spotlight)

**Comment:**

The submission introduces a novel framework for surgical video-language pretraining, which enhances language supervision by integrating surgical concepts refined through large language models (LLMs). The paper presents extensive experiments across multiple surgical procedures and provides a thorough evaluation that underscores the framework's effectiveness. A key novelty is the use of LLMs to augment surgical video text descriptions, addressing the common issue of limited text narration, particularly for key frames. This approach significantly improves the quality and diversity of the generated surgical text.

The reviewers unanimously recommend accepting the paper, particularly after the authors' rebuttal addressed initial concerns. Therefore, I recommend acceptance of this paper.

The authors are encouraged to incorporate the discussion points and clarifications raised during the review process into the final version. Additionally, it is essential that the authors make the training dataset available by sharing links to the videos used in the study, along with the necessary preprocessing code to generate video-text pairs for training. The source of the dataset and the associated license should be clearly stated to ensure transparency and reproducibility.